# Nanohole-boosted electron transport between nanomaterials and bacteria as a concept for nano–bio interactions

Tonglei Shi[1,2], Xuan Hou[1,2], Shuqing Guo[1], Lei Zhang[1], Changhong Wei[1], Ting Peng[1] & Xiangang Hu [1✉]

Biofilms contribute to bacterial infection and drug resistance and are a serious threat to global human health. Antibacterial nanomaterials have attracted considerable attention, but the inhibition of biofilms remains a major challenge. Herein, we propose a nanohole-boosted electron transport (NBET) antibiofilm concept. Unlike known antibacterial mechanisms (e.g., reactive oxygen species production and cell membrane damage), nanoholes with atomic vacancies and biofilms serve as electronic donors and receptors, respectively, and thus boost the high electron transport capacity between nanomaterials and biofilms. Electron transport effectively destroys the critical components (proteins, intercellularly adhered polysaccharides and extracellular DNA) of biofilms, and the nanoholes also significantly downregulate the expression of genes related to biofilm formation. The anti-infection capacity is thoroughly verified both in vitro (human cells) and in vivo (rat ocular and mouse intestinal infection models), and the nanohole-enabled nanomaterials are found to be highly biocompatible. Importantly, compared with typical antibiotics, nanomaterials are nonresistant and thereby exhibit high potential for use in various applications. As a proof-of-principle demonstration, these findings hold promise for the use of NBET in treatments for pathogenic bacterial infection and antibiotic drug resistance.

[1] Key Laboratory of Pollution Processes and Environmental Criteria (Ministry of Education)/Tianjin Key Laboratory of Environmental Remediation and Pollution Control, College of Environmental Science and Engineering, Nankai University, 300350 Tianjin, China. [2] These authors contributed equally: Tonglei Shi, Xuan Hou. ✉email: huxiangang@nankai.edu.cn

Bacterial resistance caused by the overuse of antibiotics causes more serious harm than cancer[1]. The World Health Organization (WHO) has reported that approximately 700,000 people die each year due to drug-resistant bacterial infections[2], and 10 million people are expected to die each year from drug-resistant bacterial infections by 2050[3]. The rapid colonization of pathogenic bacteria and large amounts of toxic metabolites cause local tissue necrosis or systemic infection[4], and the design of approaches for effectively combating pathogenic bacteria is a tricky global problem[5,6]. Unfortunately, natural antimicrobials present unstable chemical properties, exert a limited antibacterial effect, and exhibit obvious drug resistance[7,8].

Recently, the development of antimicrobial nanomaterials has attracted much attention, and the production of classic reactive oxygen species (ROS) or the penetration of the bacterial cell walls or membranes is considered the main mechanisms[9]. In contrast, extracellular polymeric substances (EPSs) adhered to the cell surfaces of biofilms would cut off nanomaterials or drugs from biofilms, reducing their antibacterial or anti-infectious efficacy[10–12]. The development of effective antibacterial materials that can destroy EPSs or biofilms with excellent biocompatibility and nonresistance is urgently needed.

Nanoholes can be easily made in two-dimensional nanomaterials and demonstrate high activity in catalysts and sensors[13]. Compared with nanohole-free nanomaterials, nanohole-rich nanomaterials present a large variety of unique features due to the presence of vacant electron sites[14]. The nanohole structure, which has strong quantum-limiting and edge effects, results in catalytic and photoelectric properties[15,16]. In this study, it was hypothesized that nanoholes can boost the electron transport between nanomaterials and bacteria to destroy EPSs and biofilms without the generation of resistance.

Herein, nanoholes were introduced to two-dimensional transition metal disulfide ($MoS_2$) nanosheets to inhibit *Staphylococcus aureus* (*S. aureus*) biofilm growth and infection (Fig. 1). By combining electrochemical behavior and biological analyses, the electron transport and redox reaction between the nanohole-enriched $MoS_2$ (NR-$MoS_2$) and biofilm were discovered. The anti-infection capacity of the materials was found to be excellent both in vitro and in vivo, and their biocompatibility was superior to that of traditional antibiotics and other antibacterial nanomaterials. Thus, nanohole-boosted electron transport (NBET) deeply inspires the design of antibacterial drugs or materials and contributes to a more in-depth understanding of antibacterial mechanisms.

## Results

**Formation of nanoholes with atomic vacancy through the facile attack of hydroxyl radicals.** Nanohole-free $MoS_2$ (NF-$MoS_2$) without obvious defects or holes exhibited an average thickness of $1.13 \pm 0.59$ nm and a uniform nanosheet morphology (Fig. 2a). Electron spin resonance (ESR) confirmed that the UV–Fenton reaction produced a large number of hydroxyl radicals ($\cdot$OH) (Supplementary Fig. 2). Additionally, many nanoholes were formed in nanohole-enriched $MoS_2$ (NR-$MoS_2$) via the facile attack of $\cdot$OH (Fig. 2a, denoted by blue arrows). Brunauer–Emmett–Teller (BET) surface area analysis showed that the mean pore diameter of the nanoholes was ~40 nm; thus, the NR-$MoS_2$ nanosheet was a mesoporous nanomaterial (Fig. 2b). Furthermore, high-resolution transmission electron microscopy (HR-TEM) images showed that the nanoholes could be described as atomic vacancies of Mo and S ($V_{x\text{Mo}+y\text{S}}$, Fig. 2c).

Raman spectroscopy confirmed the formation of nanoholes in NR-$MoS_2$ (Fig. 2d). Specifically, NF-$MoS_2$ exhibited two peaks at 739 and 1003 $cm^{-1}$, which were attributed to the out-of-plane Mo–S mode ($A_{1g}$) and the oscillating frequency of Mo–S, respectively[17]. In contrast, the analysis of NR-$MoS_2$ showed that the peak at 739 $cm^{-1}$ redshifted to 815 $cm^{-1}$ and that the peak at 1003 $cm^{-1}$ blueshifted to 990 $cm^{-1}$ due to the fluctuation of the out-of-plane Mo–S mode and the abundance of nanoholes in the nanosheets[18]. Positron annihilation lifetime spectroscopy (PALS) was further utilized to understand the content of the nanoholes (Fig. 2e and Table 1). The variables $\tau_{av}$, $e_1$, and $\tau_2$ were significantly increased from 302.6, 169.0, and 373.7 ps for NF-$MoS_2$ to 347.3, 189.0, and 408.4 ps for NR-$MoS_2$, respectively, confirming an increase in the nanohole status. The $\tau_1$ relative intensities ($I_1$) decreased from 53.94% to 45.90%, and the $\tau_2$ relative intensities ($I_2$) increased from 44.03% to 51.94%, which suggested that vacancy clusters played an important role in the

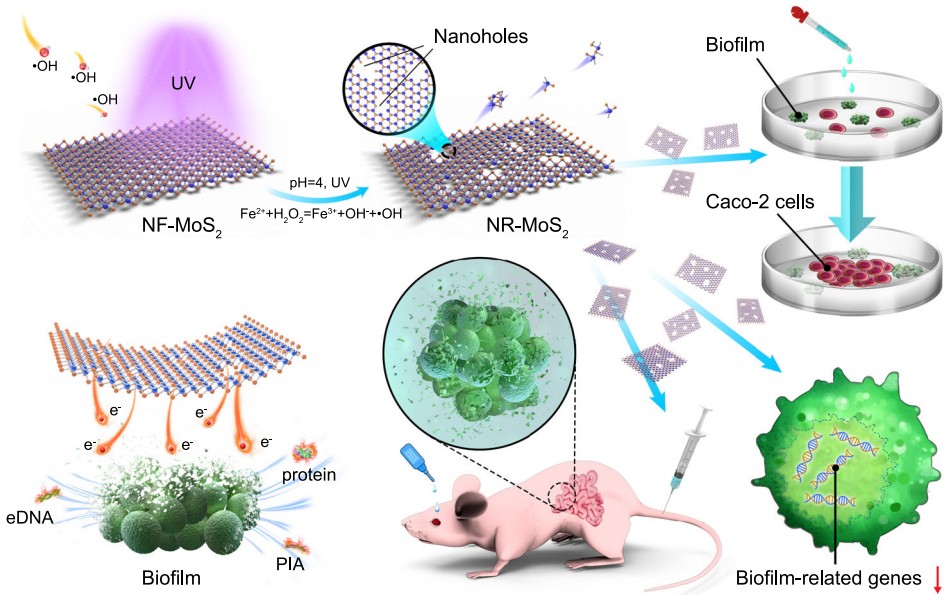

**Fig. 1 Preparation of nanohole-enriched MoS$_2$ nanosheets and the proposed anti-biofilm and anti-infection mechanisms.** NF-MoS$_2$ nanohole-free MoS$_2$, NR-MoS$_2$ nanohole-enriched MoS$_2$, eDNA extracellular DNA, PIA polysaccharide intercellular adhesin.

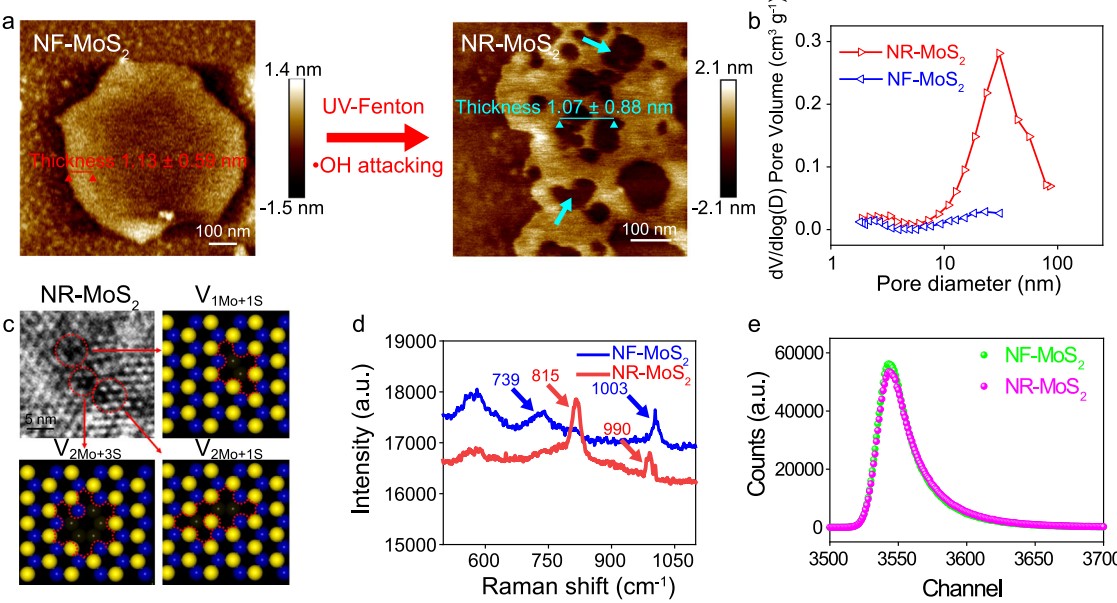

**Fig. 2 Characterization of nanohole formation with atomic vacancies in monolayer MoS₂. a** Representative atomic force microscopy (AFM) observation of NF-MoS₂ and NR-MoS₂. **b** Pore diameter distribution of NF-MoS₂ and NR-MoS₂. **c** Representative high-resolution transmission electron microscopy (HR-TEM) imaging of NR-MoS₂ and simulations of the Mo and S vacancies. **d** Raman spectra. **e** Positron annihilation lifetime spectroscopy (PALS) patterns. Representative micrographs in panels **a** and **c** were selected from three independent samples. Source data are provided as a Source Data file.

**Table 1 Position lifetime parameters of NF-MoS₂ and NR-MoS₂.**

| Sample | $\tau_1$ (ps) | $I_1$ (%) | $\tau_2$ (ps) | $I_2$ (%) | $\tau_3$ (ps) | $I_3$ (%) | $\tau_{av}$ (ps) |
|---|---|---|---|---|---|---|---|
| NF-MoS₂ | 169.4 ± 2.6 | 53.9 ± 1.0 | 373.7 ± 4.5 | 44.0 ± 0.9 | 211.8 ± 35 | 2.0 ± 0.1 | 302.6 |
| NR-MoS₂ | 189.7 ± 2.8 | 45.9 ± 0.7 | 408.4 ± 2.1 | 51.9 ± 0.7 | 222.9 ± 21 | 2.2 ± 0.1 | 347.3 |

nanoholes of NR-MoS₂. The ratios of Mo to S in NF-MoS₂ and NR-MoS₂ were 1:2.2 and 1:2.9, respectively, demonstrating that the loss of S was higher than that of Mo in the nanoholes (Supplementary Fig. 2). The X-ray photoelectron spectroscopy (XPS) results explained the increase in the Mo/S stoichiometric ratios and the formation of nanoholes.

**The high electron transport capacity of NBET contributes to the antibiofilm effect.** To optimize the nanomaterial concentration for biofilm inhibition, biofilms grown for 24, 48, and 72 h were cocultured with 0, 0.5, 1, 2, 4, 8, and 10 µg mL⁻¹ MoS₂ nanosheets. The optimized concentration of the MoS₂ nanosheets for biofilm inhibition (growth for 24, 48, and 72 h) was 4 µg mL⁻¹ (Supplementary Fig. 3a–c). NR-MoS₂ exhibited more obvious biofilm inhibition at various times (growth for 24, 48, and 72 h) than NF-MoS₂ (Supplementary Fig. 3). The biofilm inhibition was obvious but incomplete (Supplementary Fig. 3d). Due to the negatively charged EPSs on the biofilm surface and the positive charges on the MoS₂ nanosheets (Supplementary Fig. 4), the MoS₂ nanosheets could effectively interact with the biofilm. The affinity response of NR-MoS₂ was 14.71 nM, and this value was 1.3-fold higher than the value of 11.44 nM obtained for pristine MoS₂ (NF-MoS₂, Fig. 3a), which indicated a high affinity at the nM (10⁻⁹) level. The surface plasmon resonance (SPR) biosensor utilized in the present work has been widely used in the medical field but has not been previously applied to nanomaterials with anti-biofilm properties. Thus, this study describes a new application of SPR biosensors in nanomaterials with anti-biofilm properties. The affinity response values corresponded to the binding strength of bioactive molecules on the sensor surface with

nanomaterials; thus, the nanoholes acted as active sites that promoted the above-described interaction. The defects in the MoS₂ lattice as active sites promoted high affinity through the binding of hydroxide and carboxyl groups to the MoS₂ nanosheet surface[19]. Density functional theory (DFT) calculations proved that the binding energy of hydroxide and carboxyl groups adsorbed to defect-rich MoS₂ was higher than that of defect-free MoS₂ nanosheets[19]. Fourier transform infrared (FTIR) and 2D-Fourier transform infrared-correlation spectra (2D-FTIR-COS) analysis also confirmed that COO⁻ symmetric stretching was the targeted group in NR-MoS₂ (Fig. 4b). The time-dependent intracellular oxidative stress levels were analyzed using the 2,7-dichlorodi-hydrofluorescein diacetate (DCFH-DA) assay. The MoS₂ nanosheets did not increase the intracellular oxidative stress levels compared with those obtained for untreated *S. aureus* (Supplementary Fig 5a, b), and the reduced oxidative activity of the biofilms was consistent with the low microbial activities observed in the study and the detected inhibition of biofilm formation (Supplementary Fig. 5c). ·OH peaks were not observed, and the level of ¹O₂ was not increased by treatment with MoS₂ nanosheets (Supplementary Fig. 5d and e), which supported the finding that the MoS₂ nanosheets did not increase the intracellular oxidative stress of cells. In Supplementary Fig. 6, the fluorescence intensity represents the hyperpolarization and depolarization process of cells. A higher fluorescence intensity at 60 min was obtained with NF-MoS₂ (red line) compared with the control, but the difference was not significant. The intracellular pH was also not significantly disordered (Supplementary Fig. 7). The above-described results indicated the potential existence of other hidden antibacterial mechanisms that inhibit biofilm

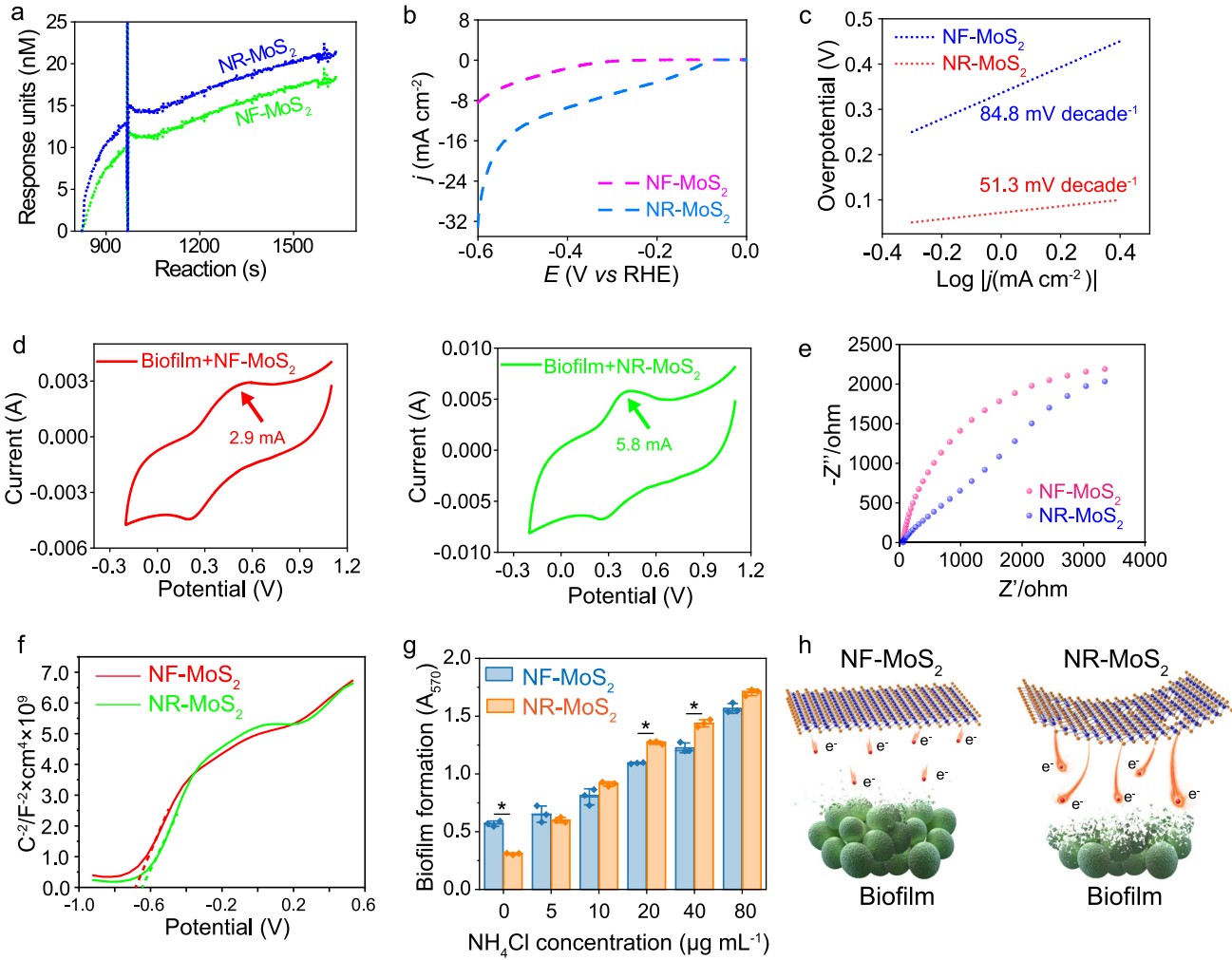

**Fig. 3 Inhibition of biofilms by nanohole-boosted electron transport. a** Affinity between the biofilm and $MoS_2$ nanosheets measured by surface plasmon resonance. **b**, **c** Linear sweep voltammetry (LSV) curves (**b**) and corresponding Tafel slopes (**c**) of $NF-MoS_2$ and $NR-MoS_2$. **d** Cyclic voltammetry (CV) curves of the $NF-MoS_2$ and $NR-MoS_2$ nanosheets. **e** Nyquist plots of $NF-MoS_2$ and $NR-MoS_2$. **f** Mott–Schottky plots of $NF-MoS_2$ and $NR-MoS_2$. **g** Effects of blocking electron transport on biofilm formation. Data represent the mean ± SD ($n = 3$ biologically independent samples). Significance was assessed using a two-sided Student's $t$-test with multiple comparisons: *$p < 0.05$. **h** Schematic diagram showing biofilm inhibition by nanohole-boosted electron transport. Source data are provided as a Source Data file.

formation rather than the well-accepted increase in intracellular oxidative stress[20,21].

Encouraged by the electrochemical catalysis of nanoholes[13–15], marked electrochemical reactions were detected between the biofilms and $MoS_2$ nanosheets. Using the normalized geometric area, the maximum dynamic current density obtained with $NR-MoS_2$ was 32.95 mA cm$^{-2}$ at an overpotential of 600 mV, which was markedly higher than that found with $NF-MoS_2$ (8.23 mA cm$^{-2}$). The corresponding Tafel slopes of $NR-MoS_2$ and $NF-MoS_2$ were 51.3 and 84.8 mV decade$^{-1}$, respectively (Fig. 3b, c). A low Tafel slope indicates more efficient electron transport kinetics at a constant overpotential[22]. Based on the cyclic voltammetry (CV) scans of the $MoS_2$ nanosheets and biofilms, the corresponding peaks were found at 2.9 and 5.8 mA (Fig. 3d), respectively, which confirmed that the biofilm was an electron acceptor and that the $MoS_2$ nanosheets serves as an electron donor. As evidenced by the electrochemical impedance spectroscopy (EIS) results, $NR-MoS_2$ with nanoholes showed lower electrochemical impedance and stronger electron transport capacity than $NF-MoS_2$ nanosheets (Fig. 3e). Mott–Schottky plots supported the finding that $NR-MoS_2$ exhibited an electron carrier density of $3.63 \times 10^{21}$ cm$^{-3}$,

which was higher than that ($3.27 \times 10^{21}$ cm$^{-3}$) of $NF-MoS_2$ (Fig. 3f). To ensure that the $MoS_2$ nanosheets were an electron donor, the generation of electrons was measured using ammonium salt, which is a classic electron quencher[23]. Ammonium salt significantly weakened the antibiofilm effect of the $MoS_2$ nanosheets, particularly $NR-MoS_2$ (Fig. 3g and Supplementary Fig. 8). As displayed in Fig. 3g, in the absence of $NH_4Cl$, the $MoS_2$ nanosheets exhibited a strong antibiofilm effect, and biofilm formation was inhibited (low $A_{570}$). The addition of ammonium salt as an electron quencher blocked the electron exchange between the biofilm and $MoS_2$ nanosheets, leading to a decreased antibiofilm effect and good biofilm formation (high $A_{570}$). At the low ammonium salt concentration of 5 μg mL$^{-1}$, the capacity of the electron quencher was low, and increases in the concentration of ammonium salt from 10 to 80 μg mL$^{-1}$ resulted in increases in biofilm formation, which supported the finding that the electron transport mechanism played an important role in antibiofilm activity. Ammonium salt at concentrations of 10–80 μg mL$^{-1}$ exerted an effect that supported the electron transport mechanism, and the findings showed that ammonium salt played a more important role in the effect of $NR-MoS_2$ than in that of $NF-MoS_2$

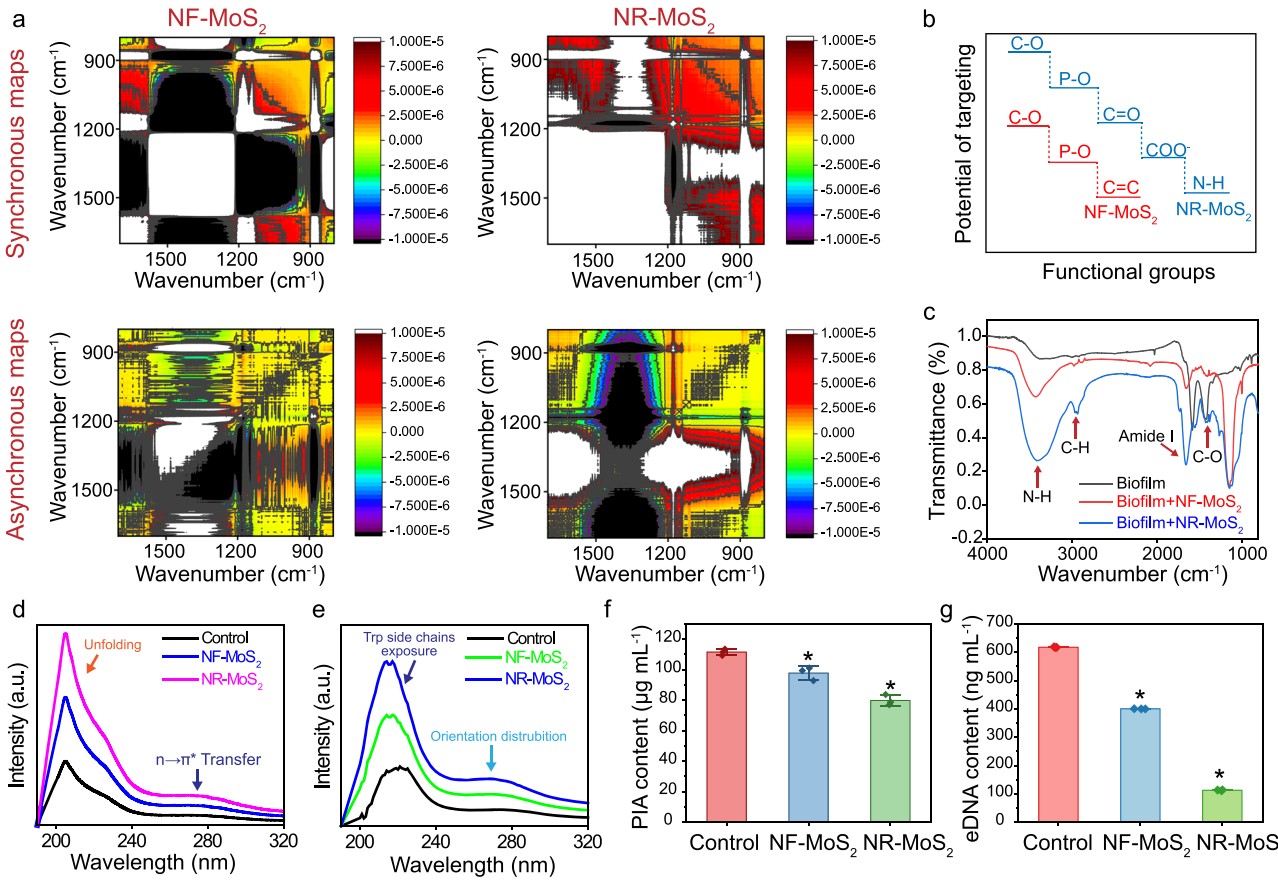

**Fig. 4 Targeting critical active sites on the biofilm by electron transport. a** Synchronous and asynchronous 2D-Fourier transform infrared-correlation spectra maps (2D-FTIR-COS) of the FTIR spectra of biofilms treated with $MoS_2$ nanosheets. **b** Potential functional groups of biofilms targeted by $MoS_2$ nanosheets. **c** FTIR spectra of the untreated biofilm (control), biofilm + NF-$MoS_2$ and biofilm + NR-$MoS_2$ samples. **d** UV–visible absorption of biofilms treated with NF-$MoS_2$ and NR-$MoS_2$. **e** UV differential spectrum of biofilms treated with NF-$MoS_2$ and NR-$MoS_2$. **f** PIA contents of the biofilms. **g** eDNA contents of the biofilms. Data represent the mean ± SD ($n = 3$ biologically independent samples). Significance was assessed using a two-sided Student's $t$-test with multiple comparisons: *$p < 0.05$. Source data are provided as a Source Data file.

at concentrations of 20 and 40 $\mu g\,mL^{-1}$. Moreover, the electrochemical characterization shown in Fig. 3b–f confirmed the better electron transport capacity of NR-$MoS_2$. Above all, the results showed that the electron transport mechanism plays an important role in the antibiofilm activity of NR-$MoS_2$. The above-described findings propose and confirm a state-of-the-art concept for the antibiofilm mechanism: the nanohole structure boosts the electron transfer efficiency and thereby reduces and inhibits biofilm formation (Fig. 3h).

**Targeting critical active sites on the biofilm by electron transport.** Proteins, polysaccharide intercellular adhesin (PIA) and extracellular DNA (eDNA) are the main active components of biofilms[22]. The electron transport targeting the critical active components was analyzed by 2D-FTIR-COS analysis, FTIR spectra, UV–visible absorption/differential spectra and detection of the biofilm components. A 2D-FTIR-COS analysis was conducted to explore the targeting of the active sites on biofilms by extending the spectra along the second dimension and discerning the relative directions and specific orders of the structural variations[24]. As illustrated in the synchronous maps shown in Fig. 4a, most of the response peaks were located at 1000–1600 $cm^{-1}$, which indicated that proteins, polysaccharides, and phosphates (nucleotide) responded to the main interfacial attack components[25]. The asynchronous map could identify specific responding functional groups at the bacteria–nanomaterial interface[25]. As provided by the asynchronous mapping (Fig. 4a), C–O stretching

of the polysaccharide, P–O stretching from phosphate and aromatic C = C stretching (1400–1500 $cm^{-1}$) were the main targeted groups in NF-$MoS_2$, whereas C–O stretching of the polysaccharide, P–O stretching from phosphate, C = O stretching, $COO^-$ symmetric stretching and N–H (amide I and II) were the targeted groups in NR-$MoS_2$. The main results presented in Fig. 4a are illustrated in Fig. 4b. After electron transport, the N–H and C–H groups in the biofilm increased, whereas the C–O group decreased (Fig. 4c). Significant changes were found in the amide I bond between the treated and control groups, which indicated that electron transport mainly attacked the proteins on the biofilm.

Compared with that of the control, the intensity of the UV–visible absorption peaks of biofilms at 204 nm gradually increased with the addition of $MoS_2$ nanosheets, which indicated that electron transport reduced the helication and unfolding of the protein skeleton (Fig. 4d). The absorption intensity of the absorption peaks of the biofilms at 226 and 286 nm increased after the addition of NR-$MoS_2$ nanosheets (Fig. 4e), demonstrating that electron transport significantly changed the orientation distribution of the peptide skeleton side chains. PIA and eDNA, as components of the biofilm, play an important role in initial bacterial adhesion to contact surfaces and act as a structural component to stabilize the cell density in an established biofilm. Both the contents of PIA and eDNA, as critical active components targeted by electron transport, were significantly reduced by the NR-$MoS_2$ nanosheets (Fig. 4f, g), and the results were consistent with the 2D-FTIR-COS and FTIR analyses. The

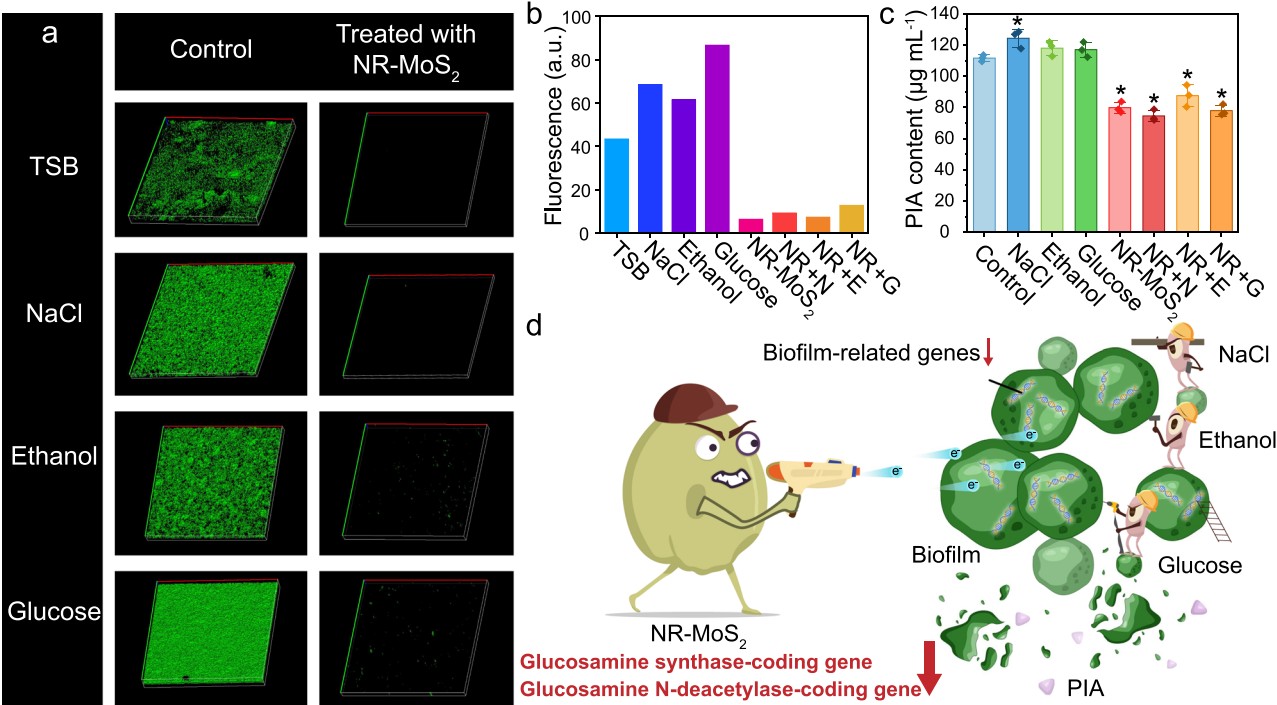

**Fig. 5 Electron transport inhibits biofilm formation and related gene expression under physiological conditions. a** Representative 3D-reconstructed confocal images of biofilms. The length and width were both 800 nm. **b** Calculated florescence intensity from 3D-reconstructed confocal images of the biofilms. The biofilms were double-stained with SYTO9 and PI. **c** PIA contents of the biofilms. The biofilm inducers glucose (G), ethanol (E), and NaCl (N) were added to the TSB medium to final concentrations of 0.5%, 4%, and 2%, respectively. The concentration of NR-MoS$_2$ (NR) was 2 µg mL$^{-1}$. Data represent the mean ± SD ($n = 3$ biologically independent samples). Significance was assessed using a two-sided Student's $t$-test with multiple comparisons: $*p < 0.05$. **d** Schematic diagram showing the NR-MoS$_2$-induced inhibition of biofilm formation under physiological conditions. Source data are provided as a Source Data file.

above-described findings supported the conclusion that electron transport targets the critical active components (e.g., proteins, PIA and eDNA) of biofilms.

**Electron transport inhibits biofilm formation and related gene expression under physiological conditions.** NaCl, glucose, and ethanol are widely present in biological fluids or during disinfection processes but are ignored in the assessment of antibacterial drugs or materials[23,26,27]. Unexpectedly, biofilm formation increased significantly with the addition of NaCl, glucose, and ethanol (Supplementary Fig. 9), and based on this finding, previous antibacterial drugs or materials deserve reconsideration[7,28].

During culture without NR-MoS$_2$, most bacteria lived in the biofilm, and NaCl, glucose, and ethanol-enhanced biofilm formation (Fig. 5a, b). The NR-MoS$_2$ treatment markedly weakened the green fluorescence in the biofilm and significantly decreased the density of the biofilm compared with that of the control and other treated groups. Slight hollows were observed on bacterial cells after NR-MoS$_2$ treatment through scanning electron microscopy (SEM) imaging and surface height topology analysis (Supplementary Fig. 10). A similar result was found with Gram-negative bacterial biofilms (*Escherichia coli*, *E. coli*). Specifically, *E. coli* biofilm formation was significantly inhibited after NR-MoS$_2$ treatment (Supplementary Fig. 11a and b). *E. coli* cells exhibited more serious hollows on the cell surface after treatment with the MoS$_2$ nanosheets than untreated cells (black arrows in Supplementary Fig. 11c). To explain the inhibition mechanisms, the expression of genes related to biofilm formation were investigated. Glucose, NaCl, and ethanol activated *Sig*B and *Rsb*U factors and induced the formation of PIA, which is an

important component of biofilms[29]. In contrast, the PIA biomass was significantly reduced by NR-MoS$_2$, even in the presence of NaCl, glucose, and ethanol (Fig. 5c). In contrast, NF-MoS$_2$ exerted a limited biofilm inhibition effect and did not reduce the PIA content (Supplementary Fig. 12). The glucosamine synthase- and glucosamine N-deacetylase-coding genes, which are involved in PIA synthesis, were also downregulated after NR-MoS$_2$ treatment (Supplementary Fig. 13). The results support the conclusion that nanoholes inhibited the expression of genes related to biofilm formation, even in the presence of NaCl, glucose, and ethanol (Fig. 5d).

**Inhibition of pathogenic bacterial adhesion and invasion of human cells.** Prior to the anti-infection experiment, the biocompatibility of NR-MoS$_2$ was investigated. After 24 h of coculture with Caco-2 cells, a negligible loss cell viability was observed in the control group (Supplementary Fig. 14a), whereas NR-MoS$_2$ significantly inhibited the adhesion and invasion of Caco-2 cells by *S. aureus* (Fig. 6a). Caspase-3 is an important effector enzyme of apoptosis[30]. *S. aureus* significantly induced Caco-2 cell apoptosis, but NR-MoS$_2$ reduced caspase-3 activity and effectively alleviated the apoptosis of intestinal epithelial cells induced by *S. aureus* (Fig. 6b). The infected intestinal tract induced the release of NO and further enhanced the inflammatory response[31,32]. Moreover, the degree of extracellular lactate dehydrogenase (LDH) leakage, which is an indicator of the membrane damage, induced by *S. aureus* was significantly mitigated by NR-MoS$_2$ (Fig. 6c). Compared with the control group, the biofilm inducers induced an ~1.2–1.7-fold increase in NO release, whereas the NO release from NR-MoS$_2$-treated cells was only one-third of the control value (Fig. 6d). The therapeutic effects and mechanisms

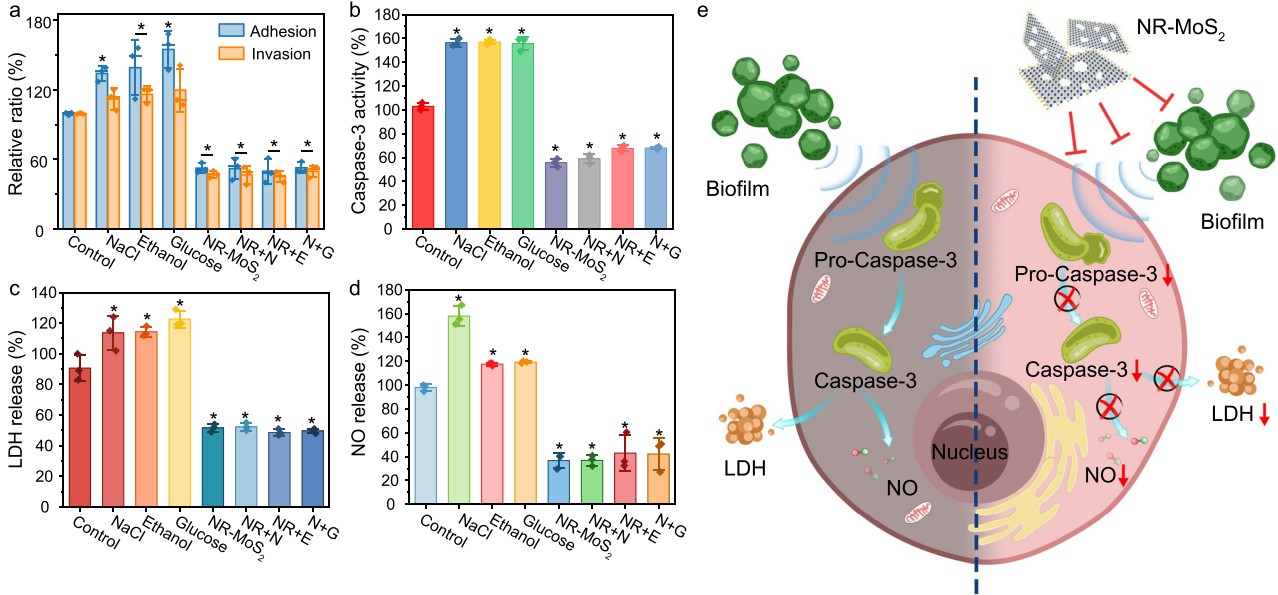

**Fig. 6 NR-MoS₂ inhibits *S. aureus* adhesion to and invasion of human cells. a** Caco-2 cells cocultured with *S. aureus*. **b** Caspase-3 activity. **c** Lactate dehydrogenase (LDH) release. **d** NO release. Data represent the mean ± SD (*n* = 3 biologically independent samples). Significance was assessed using a two-sided Student's *t*-test with multiple comparisons: *$p < 0.05$. **e** Schematic diagram showing the NR-MoS₂-induced inhibition of *S. aureus* adhesion to and invasion of human cells. The biofilm inducers glucose (G), ethanol (E), and NaCl (N) were added to the TSB medium to final concentrations of 0.5%, 4%, and 2%, respectively. The concentration of NR-MoS₂ (NR) was 2 µg mL⁻¹. Source data are provided as a Source Data file.

of NR-MoS₂ on intestinal epithelial cell injury induced by *S. aureus* are illustrated in Fig. 6e. In addition, NF-MoS₂ could not inhibit the adhesion and invasion of Caco-2 cells induced by *S. aureus* (Supplementary Fig. 14). The above-described results indicated that nanoholes can inhibit pathogenic bacterial adhesion to and invasion of human cells.

**Anti-infection effect in the animal intestinal and ocular models.** The therapeutic effect of NR-MoS₂ on murine *S. aureus* intestinal infection was efficacious, and a significant reduction in biofilm formation was observed (Fig. 7a, b). *S. aureus* infection increased the translation levels of proinflammatory cytokines (IL-1 and IL-6) by approximately 30% and 20%, respectively (Supplementary Fig. 15), but treatment with NR-MoS₂ significantly reduced their translation to the control levels. The numbers of CD11c-positive cells after treatment with NR-MoS₂ were close to those found in the control group, which was consistent with the ELISA data (Fig. 6c and Supplementary Fig. 16). Compared with severely infected tissues after local *S. aureus* infection, the intestinal mucosa was not obviously exfoliated (as denoted by the red arrows) after NR-MoS₂ treatment, and the intestinal villi were tightly rearranged (as denoted by green arrows, Fig. 7d). Moreover, the histological analysis (Fig. 7d) demonstrated the good biocompatibility of NR-MoS₂, and no obvious tissue lesions were detected after NR-MoS₂ insertion in vivo.

The anti-infection roles of NR-MoS₂ in the ocular wounds of rats treated with *S. aureus* were also tested. The infected groups displayed severe ocular swelling and corneitis (Fig. 7a). In contrast, NR-MoS₂ significantly accelerated wound healing and decreased keratitis, and the state of the eye returned to normal after 3 days of treatment (Fig. 7e). The plate-colony results confirmed that treatment with NR-MoS₂ exerted an effective anti-infection effect (Fig. 7f). The restoration of the visual sensitivity and sight of the cornea was examined using a visual water test. Compared with the infected groups, the NR-MoS₂-treated rats presented strongly enhanced visual sensitivity and markedly

improved sight during the test (Fig. 7g–i). The NR-MoS₂-treated groups spent more time in the target quadrant and more easily found the target platform than the corneal-infected groups. The rats in the NR-MoS₂-treated groups exhibited smooth and purposeful swimming behavior, and their motion paths were focused on the target quadrant during the test time, which indicated that NR-MoS₂ treatment resulted in the successful restoration of visual sensitivity and sight in the corneal-infected rat model.

**Nonresistance compared with antibiotic drug resistance.** The above-described results verified that NR-MoS₂ exhibited super capacity in anti-biofilms, as illustrated in Fig. 8a. The drug resistance of antibiotics is a serious global issue[33,34]. The resistance of *S. aureus* to NR-MoS₂ was compared with that to three typical clinical antibiotics. As shown in Fig. 8b, *S. aureus* rapidly developed resistance to the three tested antibiotics (ciprofloxacin, methicillin, and tetracycline), but resistance to NR-MoS₂ did not develop even after 20 successive generations. After 10 successive generations, the biofilm inhibitory concentrations (BICs) of the three classical antibiotics increased by 130-, 268-, and 125-fold, respectively (Fig. 8c). In contrast, *S. aureus* remained extremely sensitive to NR-MoS₂ after 20 treatments with NR-MoS₂. The results were consistent with the protein–protein interaction (PPI) analyses of the drug resistance-related cell membrane sensor and exoprotein synthesis (Fig. 8d). The main mechanism of *S. aureus* resistance to disinfectants and fungicides involve multidrug transporters located on the cell membrane expelling antimicrobial drugs out of the cell. One of the main multidrug transporters is the QacA multidrug efflux pump, which is encoded by the *qacA* gene (Fig. 9a). In the QacA system, the secondary structural units of proteins are mostly in the "α-helix" conformation and occasionally in the "3-helix" conformation (Fig. 9b). Compared with the NF-MoS₂ system, the "β-bridge" disappeared in the NR-MoS₂ system (Fig. 9c, d). When NR-MoS₂ was added, the "β-bridge" in QacA no longer appeared, but the "5-helix" conformation

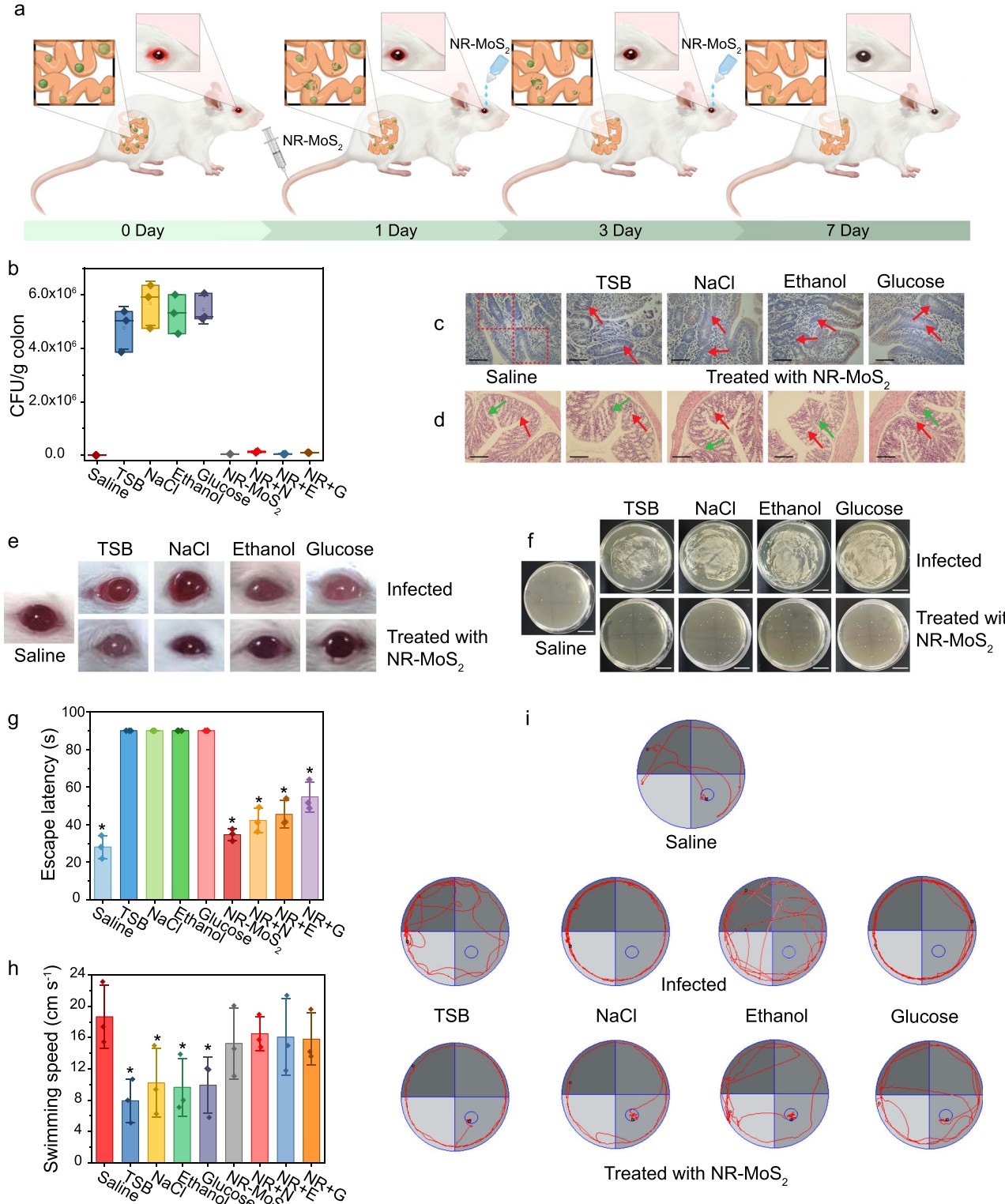

**Fig. 7 Evaluation of the anti-infection effect of NR-MoS$_2$ in the intestinal and corneal infection models. a** Schematic diagram of the evaluation of the anti-infection effects of NR-MoS$_2$ in the intestinal and corneal infection models. **b** *S. aureus* colon. **c** Analysis of CD11c immunohistochemical staining. **d** Histological hematoxylin and eosin (H&E) tissue staining analysis. "TSB" represents tryptic soy broth medium. Scale bars = 100 μm. The data were collected from eight mice per group. **e** Photographs of the ocular state of the rats during NR-MoS$_2$ treatment. **f** *S. aureus* separated from corneal tissues cultured on agar plates. Scale bars = 20 mm. **g** Escape latency in the visual water test. **h** Swimming speed in the visual water test. **i** Path tracings during the visual water test. Data represent the mean ± SD ($n = 3$ biologically independent samples). Significance was assessed using a two-sided Student's *t*-test with multiple comparisons: *$p < 0.05$. Source data are provided as a Source Data file.

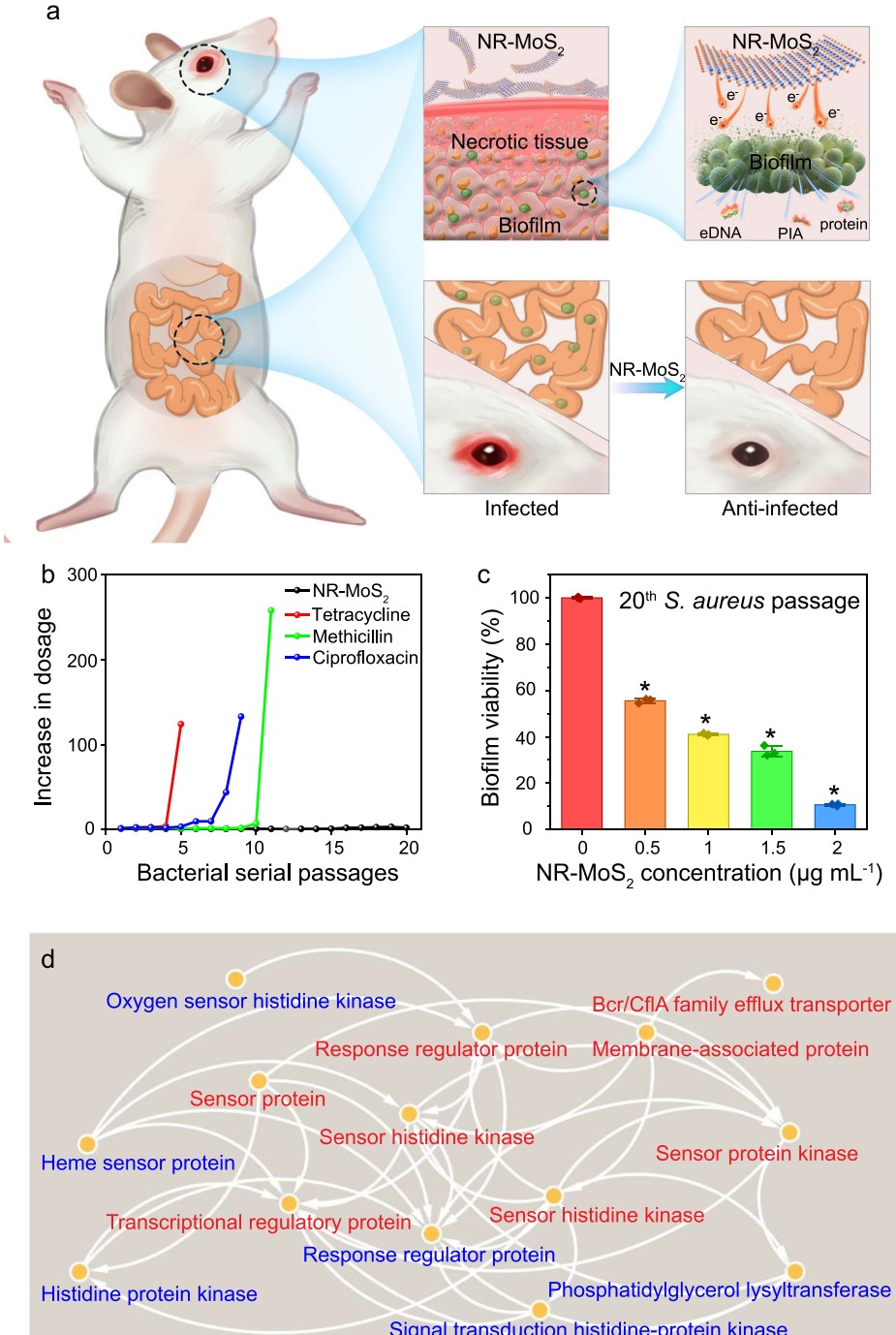

**Fig. 8 Super anti-biofilm formation in vivo and drug resistance assay. a** Diagram of super anti-biofilm formation in vivo. **b** Development of resistant *S. aureus* cells during continuous passage with a subminimum dose of bacteriostatic agents. **c** Biofilm viability after 20 serial passages measured by crystal violet staining. Data represent the mean ± SD ($n = 3$ biologically independent samples). Significance was assessed using a two-sided Student's *t*-test with multiple comparisons: *$p < 0.05$. **d** Protein–protein interaction analysis of the drug resistance-related proteins of *S. aureus* after NR-MoS$_2$ treatment. The red and blue fonts represent downregulated and upregulated proteins, respectively. Source data are provided as a Source Data file.

appeared (Fig. 9e). This abnormal structure of QacA would lead to a reduction in bacterial resistance[35] and further supported the enhanced nondrug resistance.

## Discussion

Two-dimensional layered materials, such as transition metal dichalcogenides (TMDs), generally have crystal imperfections. TMDs consist of three atomic layers of transition metal layers sandwiched between two chalcogen layers. The structural defects in TMDs are more complicated than the simple monovacancy in graphene[36]. Compared with graphene, TMDs exhibit versatile chemistry (such as insulators, semiconductors, semimetals, and true metals), which offers diverse opportunities[37]. Our recent work utilized a Bi$_2$WO$_6$/WS$_{2-x}$ nanosheet with sulfur vacancies as a broad-spectrum bactericide and found that the nanosheet exhibited sulfur vacancies that enhanced both the electron

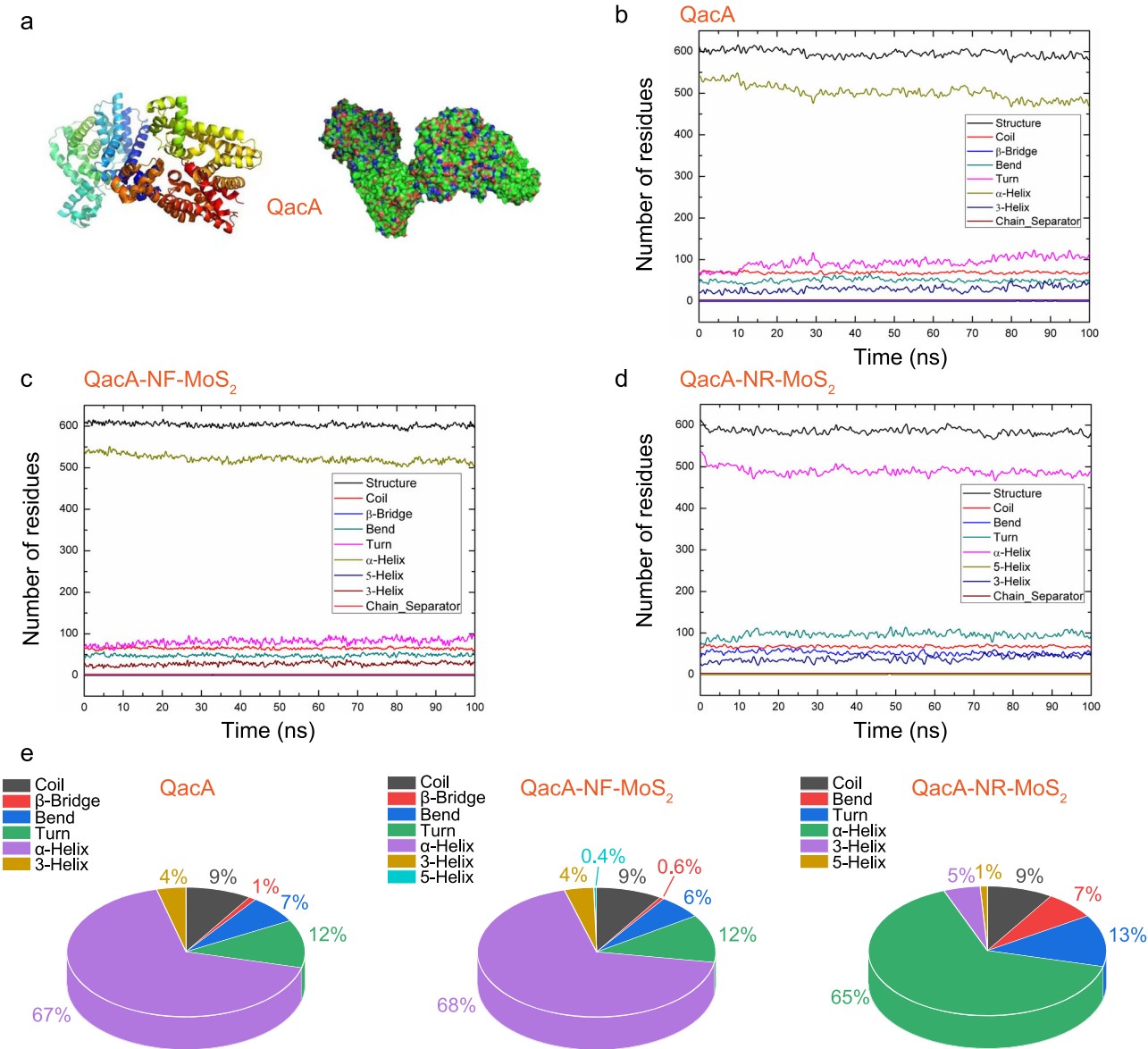

**Fig. 9 Structural alterations in the multidrug efflux pump QacA. a** Crystal structure of the membrane efflux protein QacA. **b–d** MD simulations of the binding pattern between QacA and MoS$_2$ nanosheets. **e** Alterations in the secondary structure of QacA after treatment with NF-MoS$_2$ and NR-MoS$_2$. Source data are provided as a Source Data file.

transport effect and the antibacterial efficiency of the material[38]. Nanoholes are facile to fabricate, have high activity and attract much attention in catalysis and electronic signal analysis, but their roles in biology, particularly their anti-biofilm and anti-infection activities, remain largely unknown. The UV-Fenton reaction generated free radicals (·OH) and then promoted the formation of nanohole-rich MoS$_2$ (NR-MoS$_2$), as shown in Fig. 2. The previous DFT results confirmed that ·OH tended to attack the Mo–S bond, resulting in the formation of a V$_{x\text{Mo}+y\text{S}}$ structure[39]. The XPS peak at 235 eV was attributed to the formation of additional molybdenum trioxide (MoO$_3$)[40,41], which was ascribed to the oxidation of a portion of low-oxidation-state Mo and O-occupied S vacancies (Supplementary Fig. 2). The present work proposed the following a concept for material design and an anti-infection mechanism: NBET between nanomaterials and bacteria, as confirmed in Fig. 3. The maximum dynamic current density and corresponding Tafel slopes of NR-MoS$_2$ and NF-MoS$_2$ indicated that NR-MoS$_2$ was a better electron donor than NF-MoS$_2$ (Fig. 3b, c). Furthermore, the CV curves showed the

occurrence of the Volmer reaction[42], which is a redox reaction process that converts electrons into biofilms on the MoS$_2$ nanosheet surface. NBET between nanomaterials and bacteria might be suitable for other nanomaterials with nanoholes, such as graphene-based nanomaterials[43].

Biofilm formation is a major challenge in antibacterial applications. Electron transport inhibited biofilm formation and related gene expression under physiological conditions (Fig. 5). Furthermore, NR-MoS$_2$ inhibited *S. aureus* adhesion to and invasion of human cells (Fig. 6). The anti-infection effects of NR-MoS$_2$ in the intestinal and corneal infection models were also evaluated (Fig. 7). The in vivo and in vitro studies confirmed that MoS$_2$ with nanoholes serves as a potential drug and inhibits the formation of biofilms. In addition, proteins, PIA, and eDNA are the main active components of biofilms[22]. The peptide skeleton plays an important role in the tertiary structure of proteins. The intensity of the UV–vis absorption peak at 274 nm gradually increased, denoting a reduction in the microenvironmental hydrophobicity of the aromatic amino acid residues in the

proteins after $MoS_2$ nanosheet treatment (Fig. 4d and e). One possible explanation for this phenomenon is that external additives disturbed the $n \rightarrow \pi^*$ electron transition in the aromatic amino acid chromophore[44]. NR-$MoS_2$ treatment downregulated the transcriptional regulator (SarA), which is the core protein of the biofilm formation process (Supplementary Figs. 13 and 17). Furthermore, accessory gene regulator protein A (AgrA, involved in EPS release), glucosamine synthase (IcaA, involved in PIA synthesis) and holin-like protein (CidA, involved in eDNA synthesis) were also downregulated after NR-$MoS_2$ treatment (Supplementary Figs. 13 and 17). Thus, these findings further confirmed that nanoholes inhibited the expression of genes related to biofilm formation, even in the presence of NaCl, glucose, and ethanol.

Clumping factors A and B and calcium-binding protein serine-aspartate repeat-containing proteins C–E are cell surface-associated proteins that promote bacterial attachment[45,46], and the genes related to these proteins were significantly downregulated by NR-$MoS_2$ treatment (Supplementary Fig. 18). Fibronectin-binding protein A, which promotes the attachment of a bacterium to the host[47], was blocked by NR-$MoS_2$ (Supplementary Fig. 18). The above-described results further indicate that nanoholes can inhibit pathogenic bacterial adhesion to and invasion of human cells at the genetic level.

Traditional antibiotics exhibit lower and lower antibacterial capacity due to drug resistance[48]. Once a drug is applied to S. aureus, the cell membrane sensor can receive the signal and transmit it to the extracellular membrane, and exoproteins can then be synthesized in large quantities to close the porin channel and reduce the permeability[49]. In addition, exoproteins are also involved in biofilm formation, and the barrier effect of biofilms is also an important reason for the emergence of drug resistance[50]. NR-$MoS_2$ downregulated the expression of genes related to cell membrane sensor and exoprotein synthesis, which resulted in making cells more permeable, blocking biofilm synthesis, and altering the structure of the main multidrug transporter QacA, and these effects led to the inhibition of drug resistance, as confirmed in Figs. 8 and 9. The use of materials with nanoholes could provide insight into the design of nonresistance drugs.

This study proposes a state-of-the-art concept and mechanism in which NBET enhances the antibiofilm and anti-infection properties of nanomaterials. Electron transport from nanoholes with atomic vacancies enhances the redox reaction with biofilms, alters the critical composition of the biofilm and the expression of related genes, and then inhibits pathogenic bacteria in vitro and in vivo. Due to its easy fabrication, excellent biocompatibility, and nonresistance compared with typical antibiotics, the concept of NBET could also be employed in other nanomaterial and antibacterial drug designs.

## Methods

**Synthesis of NR-$MoS_2$ nanosheets**. Nanohole-free $MoS_2$ (NF-$MoS_2$) nanosheets were obtained from XFNANO Materials Tech (Nanjing, China). NR-$MoS_2$ nanosheets were synthesized via a UV-Fenton reaction. The working power of the UV lamp was 150 W at 365 nm. $FeSO_4$ at 1 mM and $H_2O_2$ at 50 mM were added to NF-$MoS_2$ (100 mg $L^{-1}$) at a volume ratio of 1:4 (4.5 mL of $FeSO_4$ and 18 mL of $H_2O_2$). Subsequently, 0.1 M HCl was used to adjust the pH to 4.0. The mixture was sealed with a film and subjected to a UV-Fenton reaction for 4 h with stirring at 80 rpm in the dark. The mixture was filtered through a 0.1-μm membrane filter, and the membrane was then freeze-dried for 48 h to collect NR-$MoS_2$.

The reaction conditions used for NR-$MoS_2$ preparation were optimized based on the following: NF-$MoS_2$ treated with UV lamp irradiation for 4 h; NF-$MoS_2$ treated with $FeSO_4$ (1 mM) and $H_2O_2$ (50 mM) at a volume ratio of 1:4 under UV lamp irradiation (150 W) for 4 h; and NF-$MoS_2$ treated with $FeSO_4$ (1 mM) and $H_2O_2$ (50 mM) at a volume ratio of 1:10 under UV lamp irradiation (150 W) for 4 h. NF-$MoS_2$ after UV-Fenton treatment (1:4, 4 h) exhibited abundant holes while maintaining its nanosheet shape (Supplementary Fig. 19), and this preparation was thus selected as the optimized protocol for NR-$MoS_2$ preparation.

**Nanomaterial characterizations**. The NF-$MoS_2$ and NR-$MoS_2$ morphologies were determined by atomic force microscopy (AFM, Bruker, Dimension Icon, USA) and HR-TEM (JEOL, JEM-2800, Japan). Raman spectra were obtained using a Raman spectrometer (TEO, SR-500I-A, USA) at 514 nm. The $MoS_2$ defect level was recorded by PALS (fast-slow coincidence ORTEC system, USA). XPS (Kratos Analytical Ltd., Axis Ultra DLD, UK) analysis was conducted to analyze the chemical valences. BET results were obtained using a multistation extended automatic fast specific surface area meter (Mac Ltd., ASAP-2460, USA). The zeta potential of the $MoS_2$ nanosheets was examined using a ZETAPALS/BI 200SM device (Brookhaven Instruments Corporation, USA).

**Hydroxyl radical measurements**. Hydroxyl radicals were determined using a spin trap of 5,5-dimethyl-1-pyrroline-N-oxide (DMPO, 10 mM) and ESR spectroscopy (MiniScope 400, Germany)[51]. The ESR measurements were performed at room temperature with a microwave frequency of 9.4 GHz and a magnetic field modulation frequency of 100 kHz[52].

**Biofilm formation**. A standard Gram-positive S. aureus 25923 strain was obtained from the American Type Culture Collection (ATCC). Ethanol and NaCl were added to tryptic soy broth (TSB) medium as biofilm inducers, and the final concentrations were 0.5%, 4%, and 2%, respectively. To form biofilms, 100 μL of an S. aureus suspension ($10^8$ CFU $mL^{-1}$) and 100 μL of TSB medium (with and without inducers) were placed in 96-well plates and cultured at 37 °C. The TSB medium was refreshed every 24 h. After 48 h, the TSB medium was removed, and the S. aureus biofilms that were attached to the 96-well plates were harvested.

**Crystal violet staining**. After the biofilms were formed, the suspensions were discarded, and 250 μL of phosphate-buffered saline (PBS) was used to wash away the free bacteria. Subsequently, the nanomaterials were diluted in TSB medium to final concentrations of 0.25, 0.5, 1.0, 2.0, and 4.0 μg $mL^{-1}$. NR-$MoS_2$ was coincubated with the biofilms for 24 h, and the suspension was then removed. After washing with PBS, a 5% crystal violet stain was added to the S. aureus biofilm for 10 min, and the dye was then removed. After rinsing with PBS, 95% ethanol was added to the biofilm to dissolve and wash the excess crystal violet[53]. The absorbance was then measured using a fluorescence microplate detector (Bio-Tek, USA) at 570 nm.

**Intracellular oxidative stress**. The fluorescent probe DCFH-DA was used to determine the intracellular oxidative stress levels after exposure to $MoS_2$ nanosheets at 10 μg $mL^{-1}$. The bacterial cells were centrifuged at 1700 × g for 5 min, washed with PBS, and resuspended in fresh PBS buffer. The cells were stained by gentle incubation with 10 μg $mL^{-1}$ DCFH-DA at 37 °C for 30 min in the dark. The cells were collected and washed with PBS buffer, and the fluorescence density was measured using a fluorescence microplate detector (Bio-Tek, USA) with an excitation wavelength of 488 nm and an emission wavelength of 520 nm. ESR was conducted to determine the specific radicals in biofilms treated with $MoS_2$ nanosheets. Hydroxyl radicals were determined using a spin trap of DMPO (200 mM), and singlet oxygen was trapped by 2,2,6,6-tetramethyl-4-piperidine (TEMP). Briefly, S. aureus biofilms resuspended in PBS were cocultured with 10 μg $mL^{-1}$ $MoS_2$ nanosheets and the spin-trapping adduct for 15 min at 37 °C. Subsequently, S. aureus was subjected to three freeze–thaw cycles and used for the ESR assay.

**XTT assay**. 2,3-Bis[2-methyloxy-4-nitro-5-sulfophenyl]−2H-tetrazolium-5-carboxanilide (XTT) solution (1 mg $mL^{-1}$) was prepared in PBS (pH 7.4), sterilized through a 0.22-μm-pore-size filter and stored at −80 °C. Menadione (Aladdin, China) solution in acetone (0.4 mM) was sterilized through a 0.22-μm-pore-size filter. S. aureus biofilms were cultured for 24 h and then treated with $MoS_2$ nanosheets at a concentration of 10 μg $mL^{-1}$. The biofilm was washed with PBS buffer (pH 7.2) and dried in the reverse position, and 180 μL of PBS and 20 μL of XTT-menadione solution (12.5 XTT:1 menadione) were then added to the wells. The mixtures were incubated at 37 °C for 2 h, and the absorbance was then measured using a fluorescence microplate detector (Bio-Tek, USA) at 492 nm.

**Cell membrane potential**. The cell membrane potential was measured using a Cell Membrane Potential Detection Kit (BB-4110, BestBio, China). BBcellProbe M09, which is the fluorescent probe used in this study, is a lipophilic anionic fluorescent dye that detects the cell membrane potential. M09 emits fluorescence only when it enters the cell and binds to proteins in the cytoplasm. Increased cellular depolarization results in the entry of higher amounts of the M09 probe and thus increases in the intracellular fluorescence intensity. Conversely, a decrease in the intracellular fluorescence intensity indicates that the cells are hyperpolarized and exhibit a lower cell membrane potential. Briefly, S. aureus was centrifuged at 1700 × g for 5 min, washed with PBS buffer (pH 7.2), resuspended in 500 μL of M09 working solution (dilution ratio: 1:2000) and incubated gently at 37 °C for 30 min in the dark. The fluorescence density was recorded using a fluorescence microplate detector (Bio-Tek, USA) with an excitation wavelength of 488 nm and an emission wavelength of 515 nm.

**Intracellular pH measurement**. *S. aureus* cells were collected and washed with 50 mM 4-(2-hydroxyethyl)-1-piperazineethanesulfonic acid (HEPES) buffer (pH 8.0), and the samples were then mixed with 3 μM carboxyfluorescein diacetate-succinimidyl ester (CFDA-SE) probes and incubated gently at 37 °C for 30 min in the dark. To remove the nonconjugation effect, the samples were eluted with 10 mM glucose and washed twice with HEPES buffer. The fluorescence density was then measured using a fluorescence microplate detector (Bio-Tek, USA) with excitation wavelengths of 440 and 490 nm and an emission wavelength of 520 nm. The standard curve was determined using a series of pH buffers (50 mM glycine, 50 mM citric acid, 50 mM NaHPO$_4$·2H$_2$O, and 50 mM KCl; pH values of 3.0, 4.0, 5.0, 6.0, 7.0, 8.0, 9.0, and 10.0), and the fluorescence intensity (FI) ratios (FI$_{Ex=440 nm, Em=520 nm}$ to FI$_{Ex=490 nm, Em=520 nm}$) were recorded.

**EPS extraction and measurements**. Bacteria have a double-layered EPS structure in which loosely bound EPSs diffuse from the tightly bound EPSs that surround the cells. Because the soluble EPSs and the outer layer consisting of loosely bound EPSs are in direct contact with the MoS$_2$ nanosheets, both the soluble EPSs and the outer layer consisting of loosely bound EPSs were extracted from the biofilm cells using a previously reported method[54–56]. Briefly, *S. aureus* was centrifuged at 5000 × *g* and 4 °C for 15 min, and the supernatants were collected. After re-centrifugation at 12,000 × *g* and 4 °C for 30 min, the supernatants were mixed with a 3-fold volume of cold ethanol and stored at 4 °C for 48 h. The supernatants were discarded, and the pellets were dialyzed (3000 Da) for 48 h. The dialysis products were subsequently freeze-dried to obtain EPSs before being sealed and stored at 4 °C.

The EPS layer was electrochemically active itself and accelerated electron transfer between the cells and extracellular electron acceptors/donors. The redox reaction with EPSs was previously detected using electrochemistry technology[56]. The electrochemical behavior was observed with a typical three-electrode system consisting of a glassy carbon electrode (4 mm in diameter), a platinum-wire counter electrode, and an Ag/AgCl reference electrode with saturated KCl (potential versus the normal hydrogen electrode, +198 mV) on a CHI760E electrochemical working station (China). The redox reaction between the MoS$_2$ nanosheets and EPSs in 0.1 M KOH solution was evaluated using a CV test. The CV electrode was a glassy carbon electrode (diameter, 0.4 cm; electrode area, 0.126 cm$^{-2}$). The samples were prepared by adding 100 μL of ethanol and 50 μL of Nafion. A 5-μL sample was dropped on the glassy carbon electrode and allowed to dry. The parameters were as follows: $E_i = -0.2$ V and $E_f = 1.1$ V with a scan rate of 5 mV/s. EIS was performed at the open circuit voltage ($V_{oc}$) at a frequency range from 100 kHz to 10 mHz and with an AC amplitude of 5 mV. Linear sweep voltammetry (LSV), Tafel tests, and Mott–Schottky plots of the MoS$_2$ nanosheets were also performed at the Voc with a scan rate of 5 mV s$^{-1}$ to obtain the current density in 0.1 M KOH. The parameters for LSV were as follows: $E_i = 0$ V and $E_f = -0.6$ V with a scan rate of 5 mV/s.

**Electron quencher assay**. An electron attractor of NH$_4$Cl at a series of different final concentrations (5, 10, 20, 40, and 80 μg mL$^{-1}$) was coincubated with the *S. aureus* suspension (10$^8$ CFU mL$^{-1}$) and 2 μg mL$^{-1}$ MoS$_2$ nanosheets at 37 °C for 24 h. The biofilm inhibition was measured by crystal violet staining.

**Liposome preparation**. Liposomes, which act as ideal candidates for membrane mimics, were formed as thin lipid membranes were gradually hydrated, and the accumulated crystal bilayers became a swelled liquid. Briefly, 30 mg of lecithin (99%, Avanti Polar Lipids, USA) was dissolved in 6 mL of a chloroform/methanol (2:1, v/v) mixture. The dissolved lecithin was topped with a thin lipid layer using a rotary evaporation apparatus (Heidolph, Germany). To remove any residual organic solvents, the layer was dried at 37 °C for 2 h in a high-vacuum oven. The lipid layer was then transferred to vials, sealed, bundled, and frozen. For lipid membrane hydration, 6 mL of preheated HBS-N (GE, Boston, MA, USA) was added to the liposomes to form multilamellar vesicles (LMVs) above the phase transition temperature. The liposome vesicles were immediately shaken and stirred violently, vortexed for 1 min and sonicated for 40 min above the phase transition temperatures. Finally, small unilamellar vesicles (SUVs) were prepared by extrusion. A syringe was used to extract 2 mL of preheated HBS-N and inject this volume into another syringe to wash the assembled extrusion apparatus (Avanti Polar Lipids, USA). The LMV suspension was filtered 11 times through a 50-nm polycarbonate filter (Avanti Polar Lipids, USA) between the extrusion apparatus, and the SUVs were then obtained.

**Assay of the interfacial interaction between the biofilms and nanosheets**. SPR assays were performed using a Biacore T200 biomolecular interaction analysis system (GE Healthcare, USA). The L1 sensor chip was balanced at room temperature for 15–30 min before use. All liquids entering the Biacore T200 system were filtered through a 0.22-μm membrane and degassed. The L1 sensor chip was cleaned with 20 mM 3-[(3-cholamidopropyl)dimethylammonio]-1-propanesulfonate (CHAPS, 99%) (10 μL min$^{-1}$) and then washed under running buffer (HBS-N, 40 μL min$^{-1}$) to clean the residual detergent. SUVs at 2 mmol L$^{-1}$ were injected until the sensor diagram became flat and the response was stable. The control flow rate and time were 10 μL min$^{-1}$ and 5 min, respectively. NaOH (50 mM, 40 μL min$^{-1}$) was injected to remove the nonfixed SUVs. Before injection of the MoS$_2$ nanomaterials, the surface of the

L1 sensor chip was completely covered with SUVs by injecting 0.1 mg mL$^{-1}$ bovine serum albumin for 5 min to offset any nonspecific binding. Finally, the MoS$_2$ nanosheets were injected at 20 μL min$^{-1}$ for 3 min in the binding experiments.

In addition, the biofilm was centrifuged at 1700 × *g* for 5 min, washed with PBS buffer, and resuspended in fresh PBS buffer. For the UV–visible absorption and UV differential spectrum analyses, the samples were analyzed at 200–700 nm using a UV–vis spectrophotometer (TU-1901, Persee, China). For the 2D-FTIR-COS analysis, the biofilm was centrifuged at 1700 × *g* for 5 min, washed with PBS buffer, and freeze-dried for 24 h. The samples were resuspended in fresh PBS buffer and measured at 800–1700 nm using a FTIR spectrophotometer (Bruker Tensor 27, Bruker, Germany). The nanomaterials were used as an external perturbation, and a series of concentration-dependent ATR-FTIR spectra (biofilms treated with MoS$_2$ nanosheets at concentrations of 0–10 μg mL$^{-1}$) were obtained. The synchronous and asynchronous maps were constructed using 2D Shige software from Kwansei Gakuin University (Tokyo, Japan).

**PIA extract and measurement**. The biofilm was washed with PBS, and 2.0 mL of 0.5 M ethylenediaminetetraacetic acid (EDTA) was then added. The biofilm was placed in a water bath at 100 °C for 10 min and centrifuged at 10,000 × *g* for 5 min, and proteinase K (20 mg mL$^{-1}$, 50 μL) was subsequently added to the supernatant. The sample was then placed in a water bath at 37 °C for 2 h. The PIA content was detected using the PIA assay kit purchased from Nanjing Jiancheng Bioengineering Institute (Nanjing, China) according to provided instructions provided with the kit.

**eDNA extraction and detection**. After biofilm formation, the suspensions were discarded, and 200 μL of PBS was added to each well. Subsequently, the biofilm was coincubated with the nanomaterials at 2.0 μg mL$^{-1}$ for 24 h. The suspension was removed, and the biofilm was washed with PBS and homogenized at 10,000 rpm for 30 s. After homogenization, 200 μL of 0.5 M EDTA was added to the biofilm, and the samples were cooled at 4 °C for 1 h. The biofilm was then suspended in 50 mM TEN buffer (10 mM Tris–HCl, pH 8.0, 10 mM EDTA, and 500 mM NaCl), transferred into precooled 1.5-mL centrifugal tubes[57,58] and centrifuged at 15,000 × *g* and 4 °C for 5 min. Subsequently, 300 μL of TE buffer (10 mM Tris–HCl and 1.0 mM EDTA, pH 8.0) and 300 μL of phenol:chloroform:isoamyl alcohol (volume ratio, 25:24:1) were added to the supernatant, and the samples were centrifuged at 15,000 × *g* and 4 °C for 10 min. The supernatants were subjected to chloroform: isoamyl alcohol (300 μL, volume ratio, 24:1) extraction. The inorganic phase of the samples was collected, and 1 mL of cold ethanol and 30 μL of sodium acetate were added to the samples. The samples were stored at −20 °C overnight and centrifuged again at 15,000 × *g* and 4 °C for 20 min. After removing the supernatant, the eDNA was washed with 70% cold ethanol and dissolved in 20 μL of TEN buffer. The eDNA content was determined using a NanoDrop 2000 spectrophotometer (Thermo Scientific, USA) and counted based on the absorbance ratio of OD$_{260}$ to OD$_{280}$.

**Confocal microscopy observation**. For *S. aureus* biofilm formation, 1 mL of the *S. aureus* suspension (10$^7$ CFU mL$^{-1}$) and 1 mL of TSB medium were placed on a confocal dish and cultured at 37 °C for 24 h. Then, 2 μg mL$^{-1}$ MoS$_2$ was added to the biofilm, and the mixture was gently mixed and cultured at 37 °C for 24 h. The biofilms were washed twice with PBS and then stained with 200 μL of LIVE/DEAD dye (BacLight Bacterial Kit, Invitrogen, USA) in the dark at room temperature for 15 min. Images were obtained using a confocal microscope (Carl Zeiss, LSM880 with Airyscan, Germany) and processed using ZEN Blue Configuration software (Carl Zeiss, Germany). *E. coli* biofilms were prepared using the same method described above.

**Surface morphology analysis**. Bacteria treated with 10 μg mL$^{-1}$ MoS$_2$ nanosheets were harvested, fixed with 2.5% glutaraldehyde overnight at 4 °C and then dehydrated using a gradient of 30%, 50%, 70%, 80%, 90%, 95% and 100% v/v ethanol in water for field emission scanning electron microscopy (FE-SEM, JSM-7800 F, Japan) imaging. Bacteria treated with 10 μg mL$^{-1}$ MoS$_2$ nanosheets were harvested and fixed with 2.5% glutaraldehyde at 4 °C for 2 h. Bacteria harvested at 4000 × *g* for 5 min were washed with ultrapure water and resuspended in ultrapure water. The bacterial suspension (20 μL) was dropped on clean mica slices and allowed to dry at room temperature. AFM images were obtained in tapping mode in air using silicon probes (Bruker, USA).

**RNA isolation, cDNA synthesis, and transcriptomics**. Before the *S. aureus* cells were harvested for RNA isolation, the samples were retreated with RNA Protect Bacteria Buffer (TaKaRa, Japan) at 4 °C for 1 h, centrifuged at 1700 × *g* for 5 min and stored at −80 °C. RNA isolation was performed using the RNAprep Pure Cell/ Bacteria Kit (TIANGEN, Beijing, China). Bacterial DNA was removed by treatment with DNase I (50 U/total RNA) at 37 °C for 1 h. Total RNA was then reverse-transcribed to cDNA using FastQuant RT Super Mix (TIANGEN, Beijing, China). The cDNA quality was determined using a NanoDrop 2000 spectrophotometer (Thermo Scientific, USA) based on the absorbance ratio of OD$_{260}$ to OD$_{280}$.

The samples used in the transcriptomics study were untreated biofilms (controls) and biofilms that were treated with the MoS$_2$ nanosheets. After biofilm formation, the suspensions were discarded, and 200 μL of PBS was added to each

well. Subsequently, the biofilms were coincubated with the nanomaterials at 2.0 µg mL$^{-1}$ nanomaterials for 24 h. The suspension was removed and washed with PBS, and the biofilms were collected. RNA-seq sequencing was performed by Majorbio Co., Ltd. (Shanghai, China) using an Illumina HiSeq 2000 platform. Sample collection and preparation were conducted according to the manufacturer's recommended protocol[21]. Briefly, RNA degradation and contamination were monitored on 1% agarose gels, and the RNA integrity was assessed using the RNA Nano 6000 Assay Kit and the Agilent Bioanalyzer 2100 system (Agilent Technologies, CA, USA). A total amount of 1.5 µg of RNA per sample was used as the input material for preparation of the RNA samples. The clustering of the index-coded samples was performed with a cBot Cluster Generation System using TruSeq PE Cluster Kit v3-cBot-HS (Illumina) according to the manufacturer's instructions. Genes with an adjusted $p$-value < 0.05 identified by DESeq were considered differentially expressed. PPI networks were constructed using the STRING database and drawn using Cytoscape 3.3.0 software (Leroy Hood Lab, USA).

**Quantitative real-time polymerase chain reaction (PCR).** Quantitative real-time PCR was used to detect the expression of biofilm-related genes. The quantitative real-time PCR analysis of the cDNA samples (100 ng) was performed using the SuperReal PreMix Plus Kit (SYBR Green, TIANGEN, Beijing, China). Quantitative real-time PCR was performed in triplicate using the IQ5 Multicolor Real-Time PCR Detection System (Bio-Rad, Hercules, CA, USA) according to the instructions provided with the kit. Primer sequences corresponding to the biofilm-related genes were obtained from a search of the Genebank (https://cipotato.org/genebankcip/), and the primers were synthesized by Beijing Dingguo Changsheng Biotechnology Co., Ltd. (China). The PCR primers are listed in Supplementary Table 1.

**Cell experiments.** Human intestinal adenocarcinoma cells (Caco-2 cells), which were obtained from Dingguo Changsheng Biotech (Beijing, China), were cultured in Dulbecco's modified Eagle medium (DMEM), 100 µg mL$^{-1}$ penicillin, and 100 µg mL$^{-1}$ streptomycin (Gibco, USA) supplemented with 10% fetal bovine serum (FBS, HyClone, Germany) at 37 °C in a humidified atmosphere containing 5% CO$_2$. The cells were passaged every 3 days.

For the nanomaterial biocompatibility evaluation, 20 µg mL$^{-1}$ nanomaterials were dispersed in DMEM at 0.1, 1.0, and 5.0 µg mL$^{-1}$. Caco-2 cells were collected, and $1 \times 10^4$ cells were plated in 96-well plates overnight. The medium was replaced with fresh DMEM (supplemented with 5% FBS). After 2 h, the medium was removed, and the nanomaterials were coincubated with the cells for 6, 12, 18, and 24 h. Subsequently, 10 µL of CCK-8 was added to each well, and the cells were cultured for 2 h. The absorbance was measured with a fluorescence microplate detector (Bio-Tek, USA) at 450 nm.

For the analysis of cell adhesion and invasion, Caco-2 cells were collected, and $1 \times 10^5$ cells were plated in 24-well plates for 24 h. S. aureus cells in the logarithmic phase of growth were diluted to $10^6$ CFU mL$^{-1}$ in DMEM. The S. aureus suspension (1 mL) was added to 24-well plates and cocultured for 2 h. Subsequently, the nanomaterials (2 µg mL$^{-1}$) were added to the 24-well plates and gently mixed for 2 h. The medium was removed, and the cells were washed with PBS. Triton X-100 (100 µL, 0.1%) was added to the 24-well plates, and the plates were cultured at 4 °C for 30 min. The samples were diluted 10-fold with PBS buffer, and 100 µL was coated on the tryptose soya agar (TSA) medium, cultured at 37 °C for 24 h and then counted. After the Caco-2 cells, S. aureus cells and nanomaterials were cocultured for 1 h, the medium was replaced with fresh DMEM (100 µg mL$^{-1}$ penicillin and 100 µg mL$^{-1}$ streptomycin) for the cell invasion experiment. The other steps were the same as those in the cell adhesion experiment.

For analysis of the release of LDH and NO and the caspase-3 activity measurement, Caco-2 cells were collected, and $1 \times 10^4$ cells were placed in 96-well plates for 24 h. S. aureus cells in the logarithmic phase of growth were diluted to $10^7$ CFU mL$^{-1}$ with DMEM. The S. aureus suspension (100 µL) was added to 96-well plates and cocultured for 2 h. Subsequently, nanomaterials (2 µg mL$^{-1}$) were added to the 96-well plates and gently mixed for 2 h. The medium was removed and washed with PBS. The cells were centrifuged at $110 \times g$ for 5 min and resuspended in fresh PBS buffer. The release of LDH and NO and the activity of caspase-3 were detected using kits according to the provided instructions. The LDH assay kits and caspase-3 activity assay kits were obtained from Nanjing Jiancheng Bioengineering Institute (Nanjing, China) and Beyotime Institute of Biotechnology (Shanghai, China). The NO content assay kit was obtained from Solarbio (Beijing, China).

**Animal experiments.** All the experiments were performed according to the Guidelines established by the Animal Care and Use Committee of Nankai University (Tianjin, China). Healthy 4-week ICR mice (SPF) were obtained from Charles River Labs (Beijing, China). To evaluate the anti-infection effect of the nanomaterials, an intestinal infection model was established in the ICR mice. Briefly, the test mice were infected with S. aureus ($10^8$ CFU mL$^{-1}$, 100 µL) via tail vein injection. After 6 h, an obvious infection effect was observed in each test mouse. The nanomaterials (4 µg mL$^{-1}$, 100 µL) were then directly injected into the test mice. After 7 days, the mice were euthanized, and the colorectal tissues were harvested for colony counting, CD11c immunohistochemistry staining and histological hematoxylin and eosin (H&E) staining analyses. For ELISA, the eyeball was

removed and centrifuged at $110 \times g$ for 10 min to obtain serum samples, and the samples were then stored at −20 °C for interleukin (IL-1, IL-6, and IL-10) analysis.

Using Sprague-Dawley (SD) rats, a wound in the cornea of the eye was creased with a corneal drill, and S. aureus ($10^8$ CFU mL$^{-1}$, 50 µL) was added to the wound. After 24 h, nanomaterials (4 µg mL$^{-1}$, 20 µL) were added to the wound three times each day for 72 h. After 72 h, a small amount of corneal secretion was dipped into a cotton swab and cultured on TSB plates. The ocular condition of the rats was photographed, and their visual activity in water was examined based on analyses of their escape latency, swimming speed and traced paths.

**Molecular dynamics (MD) simulations.** All the all-atom MD simulations were based on a general AMBER force field with the RESP charges and performed using the Gromacs-4.6.7 software package. The system was maintained in a relaxed liquid configuration at 310 K and 1 bar. A 2-ns particle number, volume and temperature (NVT) relaxation run and a 2-ns particle number, pressure, and temperature (NPT) relaxation run were performed before the production simulation. For the equilibrium MD simulations, the time step was 2 fs, and the total run time was 100 ns. The relaxed system was used as the starting configuration. Energy minimization was performed with a composite protocol of steepest descent using termination gradients at 100 kJ mol$^{-1}$ nm$^{-1}$. The Nose´–Hoover thermostat was used to maintain the equilibrium temperature at 310 K, and periodic boundary conditions were imposed on all three dimensions. The particle mesh-Ewald method was used to compute the long-range electrostatics within a relative tolerance of $1 \times 10^{-6}$. A cut-off distance of 1 nm was applied to the real-space Ewald interactions. The same value was used for van der Waals interactions. The library of integrated network-based cellular signatures (LINCS) algorithm was applied to constrain the bond lengths of hydrogen atoms. A leap-frog algorithm with a time step of 2 fs was used.

**Statistical analysis.** All the experiments were performed in triplicate biological independent samples, and eight replicates of the animal experiments were performed in parallel. All the data were analyzed using GraphPad Prism software (GraphPad, USA) and are presented as the means ± standard errors. Comparisons between different groups were performed with Student's $t$-test. For all tests, $p < 0.05$ was considered to indicate statistical significance.

**Reporting summary.** Further information on research design is available in the Nature Research Reporting Summary linked to this article.

## Data availability
The authors declare that all relevant data are available within the paper on reasonable request. Source data are provided with this paper.

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

## Acknowledgements

This work was financially supported by the National Natural Science Foundation of China (grant no. 21722703) and the Ministry of Education of China (grant no. T2017002).

## Author contributions

X. Hu conceived and designed the experiments. T.S. and X. Hou performed the experiments and wrote the paper. T.S., S.G., L.Z., C. Wei, T.P. contributed to data analysis. T.S. created all schematic diagrams. X. Hu contributed to the improvement of the manuscript.

## Competing interests

The authors declare no competing interest.
