## [Peer Review File · Nature Communications]

Reviewer #1 (Remarks to the Author):

The manuscript "Nanohole-boosted Electron Transport between Nanomaterials and Bacteria: A Concept for Material Design and an Anti-infection Mechanism". The idea of the study is interesting; however, it seems to me the authors tried to do too many things and not all the experiments were well planned and done. There were some experiments that were not appropriate to help the authors draw the conclusions that they drew in their study. Some of the results are questionable because of the type of experiment done. I don't think the study is of high quality enough for acceptance for publication in Nature Communications. Below, you will find a more detailed assessment of the manuscript.

1) How statistically significant was the affinity response related to NR-MoS₂ and NF-MoS₂? The values 14.71 and 11.44 nM doesn't seem very large to have a very significant difference. How this value compares to other nanomaterials in the literature with anti-biofilm properties?

2) The authors claim that the biofilm was inhibited. They tested until 4 ug/mL (supplementary Figure 3). Which still showed some biofilm growth. Was there a concentration that showed complete inhibition of the biofilm? What was the concentration? The authors should report the best concentration inhibiting completely. Also, how long was the biofilm growth in this data? 12h, 24h, 48h, 72h? The time for biofilm growth is very important and should be reported in the figure. More detailed information in the legend would help. Also, it would be good for the authors to have data for the different concentrations at different time periods. To see if it is consistent in any of the biofilm conditions tested.

3) In the text and in supplementary figure 5. The authors mentioned the levels of ROS. It is not clear in the legend which ROS was tested here? There are different reactive oxygen species that a nanomaterial can produce. The authors cannot generalize that it doesn't produce ROS by testing only one species of ROS. Please clarify in the legend and modify the statement in the text, to be more specific on which ROS the authors are referring to (line 111).

"MoS₂ nanosheets did not promote ROS production (Supplementary Figure 5)"

4) The Ammonium salt experiment discussed in line 130 only show that it significantly weakened the biofilm effect for two concentrations (at 20 and 40) but not the others (5, 10 and 80). Not sure if the authors can conclude that it "significantly weakened the antibiofilm effect to the NoS₂ nanosheets and that it confirms the electron transport mechanism". Why just these two concentrations showed an effect, but not the others? The authors should explain better before jumping into such conclusion.

5) In line 163, it is not clear how the electron transport targeted the critical active components? What kind of reaction are you trying to describe here? Figure 3C is very hard to read, should be increased. Are the measurement of the biofilm before and after exposure? Or are different samples of biofilm observed in the FTIR. Because all 3 spectra look very different, including the intensity of the peaks for the two nanomaterials and the control.

6) In figure 4 b and c, the author should also include the results of NF since it is one of the controls for NR. It is good to have this data too. Same for figure 5, the authors should include the data for NF. Could be that just MoS₂ would work well without needing the holes that the authors are claiming.

Methods section:

Overall, the materials and methods section is very disorganized. It doesn't seem to follow a specific flow. Some materials and methods that are related, such as EPS extraction and measurements, should be together and not separate since they are related or MBEC assay and crystal violet staining, RNA isolation should be together with transcriptomics since it was done to do transcriptomics, etc... It is very fragmented and hard to follow what was done with what to get what results? This section needs to be completely reorganized. Other specific comments can be found below:

7) In the experimental method, why the concentration and ratio of 1:4 for the UV-Fenton reaction was used (pg. 11 lines 325 to 333). Were these the optimum conditions? There was no previous study cited for this, so I am wondering if the authors did any optimization to determine these conditions to produce the nanoholes? If yes, the authors should show the results of their optimization in the supporting information.

8) The authors should cite the paper that describes the Hydroxyl radical measurements using the spin trap. This is not a novel method; therefore the appropriate reference should be cited.

9) The authors did the MBEC assay after the biofilm was formed. However, what would be the

biofilm inhibition before the biofilm was formed? Meaning, growing the planktonic cells with the nanomaterials?

10) Again, the crystal violet staining is not a new technique, the authors should cite the authors that developed the technique for the crystal violet staining.

11) The method described for the electrochemical characterization is not really clear. How did the authors prepare the EPS to measure electrochemical behavior? Did they use the soluble EPS or the one attached to the biofilms? Did the authors extract the EPS from the ones attached to the biofilms? More explanation on the type and experimental setup is necessary for the EPS electrochemical characterization. This reviewer is not convinced that the authors can really measure accurately the redox reaction with the EPS using this approach. Are there other works that have shown that this method works?

12) I am not seeing the rationale for measuring the electron absorption. There was no significant results. What did the authors aimed to achieve by measuring it?

13) The extracellular DNA (eDNA) detection seems to me that the authors are extracting the total DNA of the biofilm since they resuspend the whole biofilm to do the extraction. The authors didn't seem to have done not only the extracellular DNA extraction with the protocol described. Hence, the protocol doesn't match the title of the section.

14) For the EPS extraction, did the authors grow the cells in rich media (it is not clear in the description of the preparation for the EPS)? If yes, the media will also contain biomolecules that will remain in the media. The only way to get real EPS from cells is to use a salt media that can be dialyzed out from the media to get only the biomolecules from the Soluble EPS secreted by the biofilm. On another note, it seems that the authors are only analyzing the soluble EPS. But, in biofilms the main problem with antibiotics is not necessarily with the soluble EPS, but with the actual attached EPS attached to the biofilm that inhibits the penetration of the drugs in the biofilm. Based on this experiment, the authors neglected the major EPS produced by microorganisms.

15) The ROS assay is for measure responses to oxidative metabolism, also used for general oxidative stress inside cells. The authors cannot extrapolate that it is to measure ROS produced by the nanomaterials, it is not accurate. This is because cells also have mechanisms of defense against ROS, they produce a lot of biomolecules to fight against ROS. This is a very indirect measure of ROS produced by nanomaterials. It would not be accurate to extrapolate that if the intracellular ROS is low, the nanomaterial is NOT producing ROS. It could be that the ROS is being transformed by the cells in their own defense mechanisms.

16) In the cell membrane potential, the method is not clear. Which fluorescence density was measured? What is this M09 working solution? What is the fluorescent probe used? Not convinced about the method used.

17) For the transcriptomics study, it is not clear to me what conditions were used? Did the authors incubate with the nanomaterials, were the biofilms extracted? Which cells were extracted (planktonic or biofilms)? What were the conditions tested and controls used? In supplementary Table1. What these genes correspond to? Also, what are the references for the primers described in this table? Or did the authors design these primers? Need a little more information.

18) For the liposomes preparation, which lipids were used? Where were they acquired?

Reviewer #2 (Remarks to the Author):

The manuscript demonstrates a simple, cost-effective, non-resistant, biocompatible anti-bacterial nanohole-enriched MoS₂ nanosheets (NR-MoS₂) as an alternative to traditional antibiotics. The authors have proposed a nanohole-boosted electron transport (NBET) antibiofilm concept along with interesting in vitro and in vivo experimental results as a proof-of-principle demonstration. The anti-biofilm efficacy of the nanohole-enriched MoS₂ nanosheets makes it a very promising nanomaterial for the gen-next applications in near future. However, few points need to be addressed by the authors before publication:

1. The meaning of the lines in page 2, line 49-51 – “However, extracellular polymeric substances (EPSs) and the quick growth of biofilms isolate nanomaterials or drugs from bacteria, reducing the antibacterial or anti-infectious efficacy” is not clear.

2. Typo error in Page 2, line 54 – “ nanomaterials” – The authors should thoroughly check the manuscript for typo and sentence construction errors.

3. What is the advantage of using 2D transition metal disulfide MoS₂ nanosheets over other 2D

transition nanomaterials in this system? Will any other 2D nanomaterial like Tungsten disulfide (WS₂) or graphene nanosheets will produce the same nanohole-boosted electron transport (NBET) effect in order to destroy biofilms? A brief discussion must be included in the manuscript addressing this point.

4. In Figure 2(a), the NR-MoS₂ shows better affinity towards biofilms compared to that of NF-MoS₂ sheets. The authors claim that it may be due to the fact that the localized nanoholes in NR-MoS₂ acts as active sites to promote better affinity. How presence of nanoholes influence the affinity of MoS₂ towards biofilms? Include a brief explanation.

5. The control data shown in the Supplementary Figure 5-7 are of untreated bacterial cells? If yes, then it should be mentioned in figure description. Why the defects in NR-MoS₂ does not contribute to ROS generation like those of the edge defects present in the reduced graphene oxide sheets, which generally serves as active sites for the ROS generation and kills bacteria? Also, include the time-dependent ROS generation detection studies using the same DCFH-DA assay.

6. Also include the XTT assay to investigate the production of superoxide radical anion in presence of both NR-MoS₂ and NF-MoS₂ sheets (at pH 7.0 in dark) with positive control.

7. In the Supplementary Figure 6, in case of NR-MoS₂ (red line), why the intensity is higher than control at reaction time of 60 min? How intensity reading is correlated with cell membrane potential measurements?

8. In Figure 3, include the Synchronous and asynchronous 2D-Fourier transform infrared-correlation spectra maps of the FTIR spectra of untreated biofilms (as control) along with treated ones.

9. In Page 7, line 195-198, the Figure 5(c) is mentioned for NO release and Figure 5(d) is mentioned for LDH release, whereas in the legend of Figure 5 its vice-versa. Please check.

10. Upon treatment with NR-MoS₂, what happens to the membrane integrity and surface morphology of the individual bacterial cells present in the biofilm? An AFM imaging along with and the Surface height topology analysis can be carried out to investigate the matter.

Thus, I suggest a major revision at this stage.

Reviewer #3 (Remarks to the Author):

This is a quite interesting research which reports the concept of using nanoholes in two-dimensional materials for antibacterial applications. The authors have used a number of experimental and simulation techniques to study the interactions between the nanohole-riched MoS₂ sheets and the bacteria/biofilms. The results are carefully analyzed and convincing. I would like to recommend the publication of the work in NC if the authors can properly address the following issues:

The direct interaction between the bacteria and the MoS₂ nanosheets should be investigated by SEM, AFM or other techniques.

Please discuss the possibility of using the present system to inactivate Gram-negative bacteria.

How to accurately exclude the role of ROS in the antibacterial activity of the present system?

Why NF-MoS₂ was not used as the control groups in the in vitro and in vivo antibacterial evaluations?

The following related review on this topic should be cited in the Introduction section: Chem.–Asian J. 2018, 13, 3378.

REVIEWER COMMENTS

Reviewer #1 (Remarks to the Author)

The manuscript “Nanohole-boosted Electron Transport between Nanomaterials and Bacteria: A Concept for Material Design and an Anti-infection Mechanism”. The idea of the study is interesting; however, it seems to me the authors tried to do too many things and not all the experiments were well planned and done. There were some experiments that were not appropriate to help the authors draw the conclusions that they drew in their study. Some of the results are questionable because of the type of experiment done. I don't think the study is of high quality enough for acceptance for publication in Nature Communications. Below, you will find a more detailed assessment of the manuscript.

Response: Thank you for all your comments. The manuscript has been carefully revised based on each of your comments, and the quality has been thoroughly improved to meet the requirements of Nature Communications, as presented below.

1) How statistically significant was the affinity response related to NR-MoS₂ and NF-MoS₂? The values 14.71 and 11.44 nM doesn't seem very large to have a very significant difference. How this value compares to other nanomaterials in the literature with anti-biofilm properties?

Response: The affinity response values correspond to the binding strength of bioactive molecules on the sensor surface with nanomaterials. The affinity response of NR-MoS₂ with atomic vacancies increased from 11.44 (that of pristine MoS₂, NF-MoS₂) to 14.71, which corresponds to 1.3-fold increase and indicates a high affinity at the nM (10^{-9}) level. The 1.3-fold increase in affinity significantly enhanced the ability to inhibit biofilms: the biofilm-inhibition ability of NR-MoS₂ was 11.4-fold higher than that of NF-MoS₂, as shown in **Figures 4a and 4b** and **Supplementary Figures 3 and 12**. The surface plasmon resonance (SPR) biosensor utilized in the present work has been widely used in the medical field but has not been previously

used for nanomaterials with anti-biofilm properties. This study thus describes a new application of SPR biosensors in nanomaterials with anti-biofilm properties. The above information has been clarified and added in **Lines 116-125**.

2) The authors claim that the biofilm was inhibited. They tested until 4 ug/mL (supplementary Figure 3). Which still showed some biofilm growth. Was there a concentration that showed complete inhibition of the biofilm? What was the concentration? The authors should report the best concentration inhibiting completely. Also, how long was the biofilm growth in this data? 12h, 24h, 48h, 72h? The time for biofilm growth is very important and should be reported in the figure. More detailed information in the legend would help. Also, it would be good for the authors to have data for the different concentrations at different time periods. To see if it is consistent in any of the biofilm conditions tested.

Response: We have clarified the indicated issues, and the related experiments have been added.

To optimize the nanomaterial concentration for biofilm inhibition, biofilms grown for 24 h, 48 h and 72 h were cocultured with 0, 0.5, 1, 2, 4, 8 and 10 $\mu\text{g mL}^{-1}$ MoS₂ nanosheets. The optimized concentration of MoS₂ nanosheets for biofilm inhibition (growth for 24 h, 48 h and 72 h) was 4 $\mu\text{g mL}^{-1}$ (**Supplementary Figures 3a-3c**). NR-MoS₂ exhibited more obvious biofilm inhibition at different times (growth for 24 h, 48 h and 72 h) than NF-MoS₂ (**Supplementary Figure 3**). The biofilm inhibition was obvious but incomplete (it is not necessary to completely inhibit biofilms to inhibit bacterial growth) (**Supplementary Figure 3d**). The above information has been clarified in **Lines 107-113**. Detailed information has also been added to the legend of **Supplementary Figure 3**.

3) In the text and in supplementary figure 5. The authors mentioned the levels of ROS. It is not clear in the legend which ROS was tested here? There are different reactive oxygen species that a nanomaterial can produce. The authors cannot generalize that it doesn't produce ROS by testing only one species of ROS. Please clarify in the legend

and modify the statement in the text, to be more specific on which ROS the authors are referring to (line 111).

“MoS₂ nanosheets did not promote ROS production (Supplementary Figure 5)”

Response: Thank you for your reminder.

“ROS” here refers to the oxidation pressure of cells induced by MoS₂ rather than one specific radical. To clarify the statement, “ROS” has been changed to “intracellular oxidative stress”, as shown in **Supplementary Figure 5a and Lines 131-146**. Furthermore, electron spin resonance (ESR) was used to detect different specific radicals ($\cdot\text{OH}$ and $^1\text{O}_2$). $\cdot\text{OH}$ peaks were not observed, and the level of $^1\text{O}_2$ was not increased after MoS₂ nanosheet treatment (**Supplementary Figures 5d and 5e**), which support the notion that the MoS₂ nanosheets did not increase the intracellular oxidative stress of cells. The membrane potentials and intracellular pH fluctuated normally and were not disordered (**Supplementary Figures 6 and 7**). The above results indicated the potential existence of other hidden antibacterial mechanisms that inhibit biofilm formation (here, an electron transport mechanism between nanomaterials and bacteria was proposed) rather than the well-accepted increase in intracellular oxidative stress. The above information has been added to the revised manuscript, as shown in **Supplementary Figure 5, 6, 7 and Lines 131-146**.

4) The Ammonium salt experiment discussed in line 130 only show that it significantly weakened the biofilm effect for two concentrations (at 20 and 40) but not the others (5, 10 and 80). Not sure if the authors can conclude that it “significantly weakened the antibiofilm effect to the MoS₂ nanosheets and that it confirms the electron transport mechanism”. Why just these two concentrations showed an effect, but not the others? The authors should explain better before jumping into such conclusion.

Response: The explanation has been added to the revised manuscript.

To ensure that MoS₂ nanosheets served as an electron donor, electron generation was measured using ammonium salt, which is a classic electron quencher. As displayed in **Figure 2g**, in the absence of NH₄Cl, the MoS₂ nanosheets exhibited a

stronger antibiofilm effect and poor biofilm formation (low A_{570}). The addition of ammonium salt as an electron quencher blocked the electron exchange between the biofilms and MoS₂ nanosheets, leading to a decreased antibiofilm effect and better biofilm formation (high A_{570}).

At a low ammonium salt concentration of 5 $\mu\text{g mL}^{-1}$, the capacity of the electron quencher is low. Increases in the ammonium salt concentrations from 10 to 80 $\mu\text{g mL}^{-1}$ resulted in increased biofilm formation, which supported the finding that the electron transport mechanism plays an important role in antibiofilm activity. The effects of the concentrations (10-80 $\mu\text{g mL}^{-1}$) of ammonium salt supported the existence of an electron transport mechanism, and ammonium salt played a more important role in the effect of NR-MoS₂ than in that of NF-MoS₂ at concentrations of 20 and 40 $\mu\text{g mL}^{-1}$. Moreover, the electrochemical characterization shown in **Figures 2b-2f** confirmed that the electron transport capacity of NR-MoS₂ was better than that of NF-MoS₂. The above information has been added to the revised manuscript, as shown in **Lines 162-181**.

5) In line 163, it is not clear how the electron transport targeted the critical active components? What kind of reaction are you trying to describe here? Figure 3C is very hard to read, should be increased. Are the measurement of the biofilm before and after exposure? Or are different samples of biofilm observed in the FTIR. Because all 3 spectra look very different, including the intensity of the peaks for the two nanomaterials and the control.

Response: The electron transport targeting the critical active components was analyzed by 2D-FTIR-COS analysis, FTIR spectra, UV-visible absorption/differential spectra and detection of the biofilm components.

The 2D-FTIR-COS analysis reflected the changes in the components of biofilms attacked by transported electrons, and this analysis (**Figure 3a**) was conducted to explore the targeting of active sites on biofilms by extending the spectra along the second dimension and discerning the relative directions and specific orders of the structural variations (**Kuramochi, H. et Al., 2019**). As illustrated in the synchronous

maps shown in **Figure 3a**, most of the response peaks were located at 1000-1600 cm^{-1} , which indicated that proteins, polysaccharides (PIA) and phosphates (nucleotide) responded to the main interfacial attack components (**Tu, Z. et al., 2018**). The asynchronous map could identify specific responding functional groups at the bacteria-nanomaterial interface (**Tu, Z. et al., 2018**). As provided by the asynchronous mapping (**Figure 3a**), C-O stretching of the polysaccharide, P-O stretching from phosphate and aromatic C=C stretching (1400-1500 cm^{-1}) were the main targeted groups in NF-MoS₂, whereas C-O stretching of the polysaccharide, P-O stretching from phosphate, C=O stretching, COO⁻ symmetric stretching and N-H (amide I and II) were the targeted groups in NR-MoS₂. The main results in **Figure 3a** are illustrated in **Figure 3b**.

The size of Figure 3c (FTIR spectra) has been increased. Untreated biofilm (black line in **Figure 3c**) and biofilm treated with NF-MoS₂ (red line) and NR-MoS₂ (blue line) were evaluated. After electron transport, the N-H and C-H groups in the biofilm increased, whereas the number of C-O groups decreased (**Figure 3c**). Significant changes in the amide I bond were found between the treated and control groups, which indicated that electron transport mainly attacked the proteins on the biofilm.

Moreover, the results from the UV-visible absorption/differential spectra analysis and the detection of the biofilm components are provided in **Figures 3d -3g**. Both the PIA and eDNA biomass, as the critical active components targeted by electron transport, were significantly reduced by the NR-MoS₂ nanosheets, and the results were consistent with the 2D-FTIR-COS and FTIR analyses (**Figures 3a-3c**).

The above information has been clarified and added in **Lines 187-206** and **216-219**.

References:

- Kuramochi, H.; Takeuchi, S.; Kamikubo, H.; Kataoka, M.; Tahara, T., Fifth-Order Time-Domain Raman Spectroscopy of Photoactive Yellow Protein for Visualizing Vibrational Coupling in Its Excited State. *Sci. Adv.* 2019, 5, 4490.
- Tu, Z.; Guday, G.; Adeli, M.; Haag, R., Multivalent Interactions between 2D Nanomaterials and Biointerfaces. *Adv. Mater.* 2018, 30, 1706709.

6) In figure 4 b and c, the author should also include the results of NF since it is one of the controls for NR. It is good to have this data too. Same for figure 5, the authors should include the data for NF. Could be that just MoS₂ would work well without needing the holes that the authors are claiming.

References: We agree with your comments. The data for NF-MoS₂ have been added in **Supplementary Figures 10, 11, 12 and 14**. The results indicated that NR-MoS₂ with nanoholes worked better (anti-biofilm) than NF-MoS₂ without nanoholes, and related information has been added in **Lines 233-240, 244-246 and 266-268**.

Methods section:

Overall, the materials and methods section is very disorganized. It doesn't seem to follow a specific flow. Some materials and methods that are related, such as EPS extraction and measurements, should be together and not separate since they are related or MBEC assay and crystal violet staining, RNA isolation should be together with transcriptomics since it was done to do transcriptomics, etc... It is very fragmented and hard to follow what was done with what to get what results? This section needs to be completely reorganized. Other specific comments can be found below:

Response: Thank you for your reminder. The materials and methods sections have been completely reorganized; for example, the extraction and measurements of EPSs have been combined (**Line 513-543**), and the descriptions of RNA isolation and transcriptomics have been organized together (**Lines 643-670**). Moreover, the materials and methods section has been revised according to each of your comments, as presented below.

7) In the experimental method, why the concentration and ratio of 1:4 for the UV-Fenton reaction was used (pg. 11 lines 325 to 333). Were these the optimum conditions? There was no previous study cited for this, so I am wondering if the authors did any optimization to determine these conditions to produce the nanoholes? If yes, the authors should show the results of their optimization in the supporting

information.

Response: The concentration and ratio of 1:4 for the UV-Fenton reaction are the optimal conditions. The optimization process has been added in **Lines 415-422**, and the data are provided in **Supplementary Figure 19**.

8) The authors should cite the paper that describes the Hydroxyl radical measurements using the spin trap. This is not a novel method; therefore the appropriate reference should be cited.

Response: According to your suggestion, related references are cited, as shown in **Lines 438 and 440**.

9) The authors did the MBEC assay after the biofilm was formed. However, what would the biofilm inhibition before the biofilm was formed? Meaning, growing the planktonic cells with the nanomaterials?

Response: Experiments with planktonic cells before the biofilm was formed have been added. Planktonic *S. aureus* cells were cocultured with $4.0 \mu\text{g mL}^{-1}$ MoS₂ for 24 h (optimized conditions) to observe biofilm formation, and the results showed that biofilm formation was significantly inhibited by NF-MoS₂ and NR-MoS₂. The above information has been added in **Supplementary Figure 3d** and **Lines 112-113**.

10) Again, the crystal violet staining is not a new technique, the authors should cite the authors that developed the technique for the crystal violet staining.

Response: According to your suggestion, related references have been cited, as shown in **Line 458**.

11) The method described for the electrochemical characterization is not really clear. How did the authors prepare the EPS to measure electrochemical behavior? Did they use the soluble EPS or the one attached to the biofilms? Did the authors extract the EPS from the ones attached to the biofilms? More explanation on the type and experimental setup is necessary for the EPS electrochemical characterization. This

reviewer is not convinced that the authors can really measure accurately the redox reaction with the EPS using this approach. Are there other works that have shown that this method works?

Response: A detailed explanation of the type and experimental setup has been added to **Lines 514-518, 525-532 and 536-543.**

Bacteria have a double-layered EPS structure in which loosely bound EPSs diffuse from the tightly bound EPSs that surround the cells (Zhao, W. et al., 2015). Loosely bound EPSs can be separated from bacterial culture by high-speed centrifugation (12,000 g). Because soluble EPSs and the outer layer that consists of loosely bound EPSs directly react with MoS₂ nanosheets, we mainly focused on soluble EPSs and the outer layer consisting of loosely bound EPSs. We extracted both soluble EPSs and the outer layer consisting of loosely bound EPSs from biofilm cells. The method used for the extraction of EPSs in our study has been widely reported (Chen, Y. et al., 2013; Zhao, W. et al., 2015), and a detailed description has been provided in **Lines 514-518.**

The redox reaction with EPSs using electrochemistry technology was previously reported (Xiao et al., 2017). These researchers detected extracellular electron transfer during growth and information exchange with external environments or other cells. The reported results suggested that the EPS layer was electrochemically active itself and accelerated electron transfer between cells and extracellular electron acceptors/donors (Xiao et al., 2017). To clarify the electrochemical analysis of EPSs, the above clarification and reference have been added in **Lines 525-532 and 536-543.**

References:

Chen, Y. P.; Zhang, P.; Guo, J. S.; Fang, F.; Gao, X.; Li, C., Functional groups characteristics of EPS in biofilm growing on different carriers. *Chemosphere* 2013, 92 (6), 633-638.

Zhao, W.; Yang, S.; Huang, Q.; Cai, P., Bacterial cell surface properties: role of loosely bound extracellular polymeric substances (LB-EPS). *Colloids Surf B Biointerfaces* 2015, 128, 600-607.

Xiao, Y.; Zhang, E.; Zhang, J.; Dai, Y.; Yang, Z.; Christensen, H. E. M.; Ulstrup, J.;

Zhao, F., Extracellular polymeric substances are transient media for microbial extracellular electron transfer. *Science Advances* 2017, 3 (7), e1700623.

12) I am not seeing the rationale for measuring the electron absorption. There was no significant results. What did the authors aimed to achieve by measuring it?

Response: The statement on electron absorption has been changed to electron quencher.

To ensure that MoS₂ nanosheets served as an electron donor, electron generation was measured using ammonium salt, which is a classic electron quencher. As displayed in **Figure 2g**, in the absence of NH₄Cl, MoS₂ exerted a strong antibiofilm effect and inhibited biofilm formation (low A₅₇₀). The addition of ammonium salt as an electron quencher blocked the electron exchange between the biofilms and MoS₂ nanosheets, leading to a decreased antibiofilm effect and better biofilm formation (high A₅₇₀). The above results proved that the addition of ammonium salt weakened the antibiofilm effect of the NR-MoS₂ nanosheets. **Supplementary Figure 8** shows that NH₄Cl treatment alone had no effect on biofilm formation ($p > 0.05$). Overall, our electron quenching experiments confirmed that the electron transport mechanism plays an important role in the antibiofilm activity of NR-MoS₂. The above information has been clarified in **Lines 162-170**.

13) The extracellular DNA (eDNA) detection seems to me that the authors are extracting the total DNA of the biofilm since they resuspend the whole biofilm to do the extraction. The authors didn't seem to have done not only the extracellular DNA extraction with the protocol described. Hence, the protocol doesn't match the title of the section.

Response: Extracellular DNA (eDNA) extraction and detection were performed in our study, and a detailed description of the method used for eDNA extraction has been added in **Lines 604-611**.

14) For the EPS extraction, did the authors grow the cells in rich media (it is not clear

in the description of the preparation for the EPS)? If yes, the media will also contain biomolecules that will remain in the media. The only way to get real EPS from cells is to use a salt media that can be dialyzed out from the media to get only the biomolecules from the Soluble EPS secreted by the biofilm. On another note, it seems that the authors are only analyzing the soluble EPS. But, in biofilms the main problem with antibiotics is not necessarily with the soluble EPS, but with the actual attached EPS attached to the biofilm that inhibits the penetration of the drugs in the biofilm. Based on this experiment, the authors neglected the major EPS produced by microorganisms.

Response: For EPS extraction, *S. aureus* biofilms were first grown in tryptone soy broth medium (rich media) and washed with ultrapure water to remove TSB (pH 7.2), similar to previous studies (Zhao, W. et al., 2015). We extracted both soluble EPSs and the outer layer consisting of loosely bound EPSs from cells. The EPS extraction method has been widely reported (Chen, Y. et al., 2013; Zhao, W. et al., 2015). Loosely bound EPSs can be separated from the bacterial culture by high-speed centrifugation (12,000 g). Bacteria have a double-layered EPS structure in which loosely bound EPSs diffuse from the tightly bound EPSs that surround cells (Zhao, W. et al., 2015). Because soluble EPSs and the outer layer consisting of loosely bound EPSs directly react with the MoS₂ nanosheets, the soluble EPSs and outer layer consisting of loosely bound EPSs were analyzed. The above explanation and a description of the method used for EPS extraction have been added in **Lines 514-524**.

References:

Chen, Y. P.; Zhang, P.; Guo, J. S.; Fang, F.; Gao, X.; Li, C., Functional groups characteristics of EPS in biofilm growing on different carriers. *Chemosphere* 2013, 92 (6), 633-8.

Zhao, W.; Yang, S.; Huang, Q.; Cai, P., Bacterial cell surface properties: role of loosely bound extracellular polymeric substances (LB-EPS). *Colloids Surf B Biointerfaces* 2015, 128, 600-607.

15) The ROS assay is for measure responses to oxidative metabolism, also used for

general oxidative stress inside cells. The authors cannot extrapolate that it is to measure ROS produced by the nanomaterials, it is not accurate. This is because cells also have mechanisms of defense against ROS, they produce a lot of biomolecules to fight against ROS. This is a very indirect measure of ROS produced by nanomaterials. It would not be accurate to extrapolate that if the intracellular ROS is low, the nanomaterial is NOT producing ROS. It could be that the ROS is being transformed by the cells in their own defense mechanisms.

Response: We agree with your comments. Herein, “ROS” refers to the intracellular oxidative stress of cells induced by MoS₂ nanosheets. The related “ROS” description has been revised to “intracellular oxidative stress”, as shown in **Supplementary Figure 5** and **Lines 131-139**. **Supplementary Figures 5a and 5b** show that the MoS₂ nanosheets did not increase the intracellular oxidative stress level. Moreover, electron spin resonance (ESR) was conducted to detect specific radicals ($\cdot\text{OH}$ and $^1\text{O}_2$). $\cdot\text{OH}$ peaks were not observed, and the level of $^1\text{O}_2$ was not increased compared with the control (**Supplementary Figures 5d and 5e**), which supported the finding that the MoS₂ nanosheets did not increase the intracellular oxidative stress of cells. The above information has been clarified in **Lines 131-139**.

16) In the cell membrane potential, the method is not clear. Which fluorescence density was measured? What is this M09 working solution? What is the fluorescent probe used? Not convinced about the method used.

Response: The cell membrane potential was measured using a Cell Membrane Potential Detection Kit (BB-4110, BestBio, China). BBcellProbe M09, which is the fluorescent probe used in our study, is a lipophilic anionic fluorescent dye that detects the cell membrane potential. M09 emits fluorescence only when it enters the cell and binds to proteins in the cytoplasm. Increased cellular depolarization results in the entry of higher amounts of the M09 probe and thus increases in the intracellular fluorescence intensity. Conversely, a decrease in the intracellular fluorescence intensity indicates that the cells are hyperpolarized and exhibit a lower cell membrane potential. The above information has been clarified and added in **Lines 487-497**.

17) For the transcriptomics study, it is not clear to me what conditions were used? Did the authors incubate with the nanomaterials, were the biofilms extracted? Which cells were extracted (planktonic or biofilms)? What were the conditions tested and controls used? In supplementary Table1. What these genes correspond to? Also, what are the references for the primers described in this table? Or did the authors design these primers? Need a little more information.

Response: The samples used in the transcriptomics study were untreated biofilms (controls) and biofilms treated with MoS₂ nanosheets. Detailed information on the transcriptomics study has been included in **Lines 653-670**. The genes related to biofilm formation were selected according to the transcriptomics results. Primer sequences for the genes were obtained from a search of the Genebank (<https://cipotato.org/genebankcip/>), and the primers were synthesized by Beijing Dingguo Changsheng Biotechnology Co., Ltd. (China). Detailed information on the primers has been added to **Lines 677-681**, and the PCR primers are listed in **Supplementary Table 1**.

18) For the liposomes preparation, which lipids were used? Where were they acquired?

Response: Liposomes were prepared from lecithin (99%, Avanti Polar Lipids, USA). A detailed description of the method has been added to the Methods section (liposome preparation), as shown in **Lines 550-552**.

Reviewer #2 (Remarks to the Author)

The manuscript demonstrates a simple, cost-effective, non-resistant, biocompatible anti-bacterial nanohole-enriched MoS₂ nanosheets (NR-MoS₂) as an alternative to traditional antibiotics. The authors have proposed a nanohole-boosted electron transport (NBET) antibiofilm concept along with interesting in vitro and in vivo

experimental results as a proof-of-principle demonstration. The anti-biofilm efficacy of the nanohole-enriched MoS₂ nanosheets makes it a very promising nanomaterial for the gen-next applications in near future. However, few points need to be addressed by the authors before publication:

Response: Thank you for all your comments. The manuscript has been revised according to each of your comments, as detailed below.

1. The meaning of the lines in page 2, line 49-51 – “However, extracellular polymeric substances (EPSs) and the quick growth of biofilms isolate nanomaterials or drugs from bacteria, reducing the antibacterial or anti-infectious efficacy” is not clear.

Response: The indicated sentence has been improved, as shown in **Lines 49-51**. Moreover, the language and wording throughout the manuscript have been checked and revised.

2. Typo error in Page 2, line 54 – “nanomaterials” – The authors should thoroughly check the manuscript for typo and sentence construction errors.

Response: The indicated typographical error has been corrected, as shown in **Line 55**. All typos throughout the manuscript have been corrected.

3. What is the advantage of using 2D transition metal disulfide MoS₂ nanosheets over other 2D transition nanomaterials in this system? Will any other 2D nanomaterial like Tungsten disulfide (WS₂) or graphene nanosheets will produce the same nanohole-boosted electron transport (NBET) effect in order to destroy biofilms? A brief discussion must be included in the manuscript addressing this point.

Response: Two-dimensional layered materials, such as transition metal dichalcogenides (TMDs), generally have crystal imperfections. TMDs consist of three atomic layers of transition metal layers sandwiched between two chalcogen layers. The structural defects in TMDs are more complicated than the simple monovacancy in graphene (**Jeong et al., 2017**). Compared with graphene, TMDs exhibit versatile chemistry (such as insulators, semiconductors, semimetals and true metals), which

offers diverse opportunities (**Chhowalla M et al., 2013**). With respect to the WS₂ nanosheets mentioned by the reviewer, our recent work utilized a Bi₂WO₆/WS_{2-x} nanosheet with sulfur vacancies as a broad-spectrum bactericide, and this nanosheet exhibited sulfur vacancies that enhanced both the electron transport effect and the antibacterial efficiency (**Hou et al., 2020**). The related discussion has been clarified and added to the manuscript, as shown in **Lines 324-333**.

References:

Jeong, H. Y.; Jin, Y.; Yun, S. J.; Zhao, J.; Baik, J.; Keum, D. H.; Lee, H. S.; Lee, Y. H., Heterogeneous defect domains in single-crystalline hexagonal WS₂. *Advanced Materials* 2017, 29 (15).

Chhowalla M, Shin HS, Eda G, Li LJ, Loh KP, Zhang H. The chemistry of two-dimensional layered transition metal dichalcogenide nanosheets. *Nat Chem* 5, 263-275 (2013).

Hou, X.; Shi, T.; Wei, C.; Zeng, H.; Hu, X.; Yan, B., A 2D-2D heterojunction Bi₂WO₆/WS_{2-x} as a broad-spectrum bactericide: Sulfur vacancies mediate the interface interactions between biology and nanomaterials. *Biomaterials* 2020, 243.

4. In Figure 2(a), the NR-MoS₂ shows better affinity towards biofilms compared to that of NF-MoS₂ sheets. The authors claim that it may be due to the fact that the localized nanoholes in NR-MoS₂ acts as active sites to promote better affinity. How presence of nanoholes influence the affinity of MoS₂ towards biofilms? Include a brief explanation.

Response: Thank you for your reminder. The defects in the MoS₂ lattice as active sites promoted high affinity through the binding of hydroxide and carboxyl groups to the MoS₂ nanosheet surface (**Omar et al., 2019**). DFT calculations also proved that the binding energy of the hydroxide and carboxyl groups adsorbed to defect-rich MoS₂ nanosheets was higher than that found with defect-free MoS₂ nanosheets (**Omar et al., 2019**). Furthermore, FTIR and 2D-FTIR-COS analyses confirmed that COO⁻ symmetric stretching was the targeted group in NR-MoS₂ (**Figure 3b**). The above explanation has been added in **Lines 125-131**.

Reference:

Omar, A. M.; Metwalli, O. I.; Saber, M. R.; Khabiri, G.; Ali, M. E. M.; Hassen, A.; Khalil, M. M. H.; Maarouf, A. A.; Khalil, A. S. G., Revealing the role of the 1T phase on the adsorption of organic dyes on MoS₂ nanosheets. *RSC Advances* **2019**, *9* (49), 28345-28356.

5. The control data shown in the Supplementary Figure 5-7 are of untreated bacterial cells? If yes, then it should be mentioned in figure description. Why the defects in NR- MoS₂ does not contribute to ROS generation like those of the edge defects present in the reduced graphene oxide sheets, which generally serves as active sites for the ROS generation and kills bacteria? Also, include the time-dependent ROS generation detection studies using the same DCFH-DA assay.

Response: The control data were obtained with untreated *S. aureus*, as has been added in **Supplementary Figures 5-7, 9 and 11**. The “ROS” description was inaccurate, and the related description has been changed to “intracellular oxidative stress”, as shown in **Lines 131-139**. The time-dependent intracellular oxidative stress levels were determined using the DCFH-DA assay, and the MoS₂ nanosheets did not increase the intracellular oxidative stress levels compared with that obtained with untreated *S. aureus* (**Supplementary Figures 5a and 5b**), as clarified in **Lines 131-136**. Moreover, electron spin resonance (ESR) was conducted to detect specific radicals ($\cdot\text{OH}$ and $^1\text{O}_2$). $\cdot\text{OH}$ peaks were not observed, and the level of $^1\text{O}_2$ was not increased after NR-MoS₂ treatment (**Supplementary Figures 5d and 5e**), which supported the finding that NR-MoS₂ did not increase the intracellular oxidative stress of cells, as shown in **Lines 136-139**.

6. Also include the XTT assay to investigate the production of superoxide radical anion in prence of both NR-MoS₂ and NF-MoS₂ sheets (at pH 7.0 in dark) with positive control.

Response: Thank you for your reminder. As described in the revised manuscript, the XTT assay was used to evaluate the oxidative activity (superoxide radical anion) of

biofilms in the presence of NR-MoS₂ and NF-MoS₂ nanosheets. A detailed description of the XTT assay protocol has been added in **Lines 476-485**. The microbial activity was measured based on the absorbance at 492 nm, as shown in **Supplementary Figure 5c**. Biofilm formation was significantly inhibited by treatment with NF-MoS₂ and, in particular, NR-MoS₂ ($p < 0.05$). A related description has been added in **Lines 134-136**.

7. In the Supplementary Figure 6, in case of NR-MoS₂ (red line), why the intensity is higher than control at reaction time of 60 min? How intensity reading is correlated with cell membrane potential measurements?

Response: In **Supplementary Figure 6**, the fluorescence intensity represents the hyperpolarization and depolarization process of the cell. A higher fluorescence intensity at 60 min was obtained with NF-MoS₂ (red line) compared with the control, but the difference was not significant. The intracellular pH was also not significantly disordered (**Supplementary Figure 7**). The above results indicated the potential existence of other hidden antibacterial mechanisms that inhibited biofilm formation rather than the well-accepted increase in intracellular oxidative stress. The above information has been clarified in **Lines 139-146**.

8. In Figure 3, include the Synchronous and asynchronous 2D-Fourier transform infrared-correlation spectra maps of the FTIR spectra of untreated biofilms (as control) along with treated ones.

Response: The synchronous and asynchronous 2D-Fourier transform infrared-correlation spectra maps of the FTIR spectra included untreated biofilms (as a control) as well as treated biofilms. In **Figure 3c**, the untreated biofilm (control, black line in **Figure 3c**) and biofilm treated with NF-MoS₂ (red line) and NR-MoS₂ (blue line) are shown. The above information has been added in **Lines 590-594**.

9. In Page 7, line 195-198, the Figure 5(c) is mentioned for NO release and Figure 5(d) is mentioned for LDH release, whereas in the legend of Figure 5 its vice-versa. Please

check.

Response: The legend of **Figure 5** has been corrected (**Lines 260-265**).

10. Upon treatment with NR-MoS₂, what happens to the membrane integrity and surface morphology of the individual bacterial cells present in the biofilm? An AFM imaging along with the Surface height topology analysis can be carried out to investigate the matter.

Response: The membrane integrity and surface morphology of bacterial cells were investigated by AFM imaging and surface height topology, as shown in **Supplementary Figures 10d-10i** and **Lines 233-235 and 637-642**.

Reviewer #3 (Remarks to the Author)

This is a quite interesting research which reports the concept of using nanoholes in two-dimensional materials for antibacterial applications. The authors have used a number of experimental and simulation techniques to study the interactions between the nanohole-riched MoS₂ sheets and the bacteria/biofilms. The results are carefully analyzed and convincing. I would like to recommend the publication of the work in NC if the authors can properly address the following issues:

Response: Thank you for all your comments. The manuscript has been revised according to each of your comments, as described below.

The direct interaction between the bacteria and the MoS₂ nanosheets should be investigated by SEM, AFM or other techniques.

Response: SEM and AFM imaging were conducted to observe the direct interaction between the bacteria and the MoS₂ nanosheets, as shown in **Supplementary Figure 10** and **Lines 233-235 and 634-642**.

Please discuss the possibility of using the present system to inactivate Gram-negative

bacteria.

Response: Thank you for your reminder. The proposed system was used to inactivate *E. coli* (gram-negative bacteria), and related results have been added to the revised manuscript and are discussed in **Supplementary Figure 11** and **Lines 235-240**.

How to accurately exclude the role of ROS in the antibacterial activity of the present system?

Response: The “ROS” description was inaccurate, and the related description has been revised to “intracellular oxidative stress”.

To exclude the antibacterial mechanism of intracellular oxidative stress, the time-dependent intracellular oxidative stress levels were measured. The results showed that the MoS₂ nanosheets did not induce obvious changes in the intracellular oxidative stress levels in *S. aureus* biofilms in comparison with those found in untreated *S. aureus* biofilms (**Supplementary Figures 5a and 5b**). Moreover, electron spin resonance (ESR) was conducted to detect radicals in the bacteria-MoS₂ system ($\cdot\text{OH}$ and $^1\text{O}_2$). $\cdot\text{OH}$ peaks were not observed, and the level of $^1\text{O}_2$ was not increased after NR-MoS₂ treatment (**Supplementary Figures 5d and 5e**). The above results indicated the potential existence of other hidden antibacterial mechanisms that inhibit biofilm formation (e.g., nanohole-boosted electron transport, as proposed in the present study) rather than the well-accepted increase in intracellular oxidative stress. The above description has been explained in **Lines 131-139** and **144-146**.

Why NF-MoS₂ was not used as the control groups in the in vitro and in vivo antibacterial evaluations?

References: The antibacterial data for NF-MoS₂ as the control group have been added in **Supplementary Figures 3, 5, 10, 11, 12 and 14**, and related information has been added in **Lines 107-113, 132-139, 233-240, 244-246 and 266-268**.

The following related review on this topic should be cited in the Introduction section:
Chem.–Asian J. 2018, 13, 3378.1

Response: The indicated reference is useful for understanding the progress in the use of 2D nanomaterials as antibacterial agents and has been cited in the Introduction section (**Lines 46-49**).

Reviewer #1 (Remarks to the Author):

The authors have addressed all the comments. I have no further comments.

Reviewer #2 (Remarks to the Author):

The authors have now addressed most of the issues raised and the manuscript can be accepted in its present form.

Reviewer #3 (Remarks to the Author):

The authors have fully addressed all my comments.

RESPONSES TO REVIEWERS' COMMENTS

Reviewer #1 (Remarks to the Author):

The authors have addressed all the comments. I have no further comments.

Response: Thank you.

Reviewer #2 (Remarks to the Author):

The authors have now addressed most of the issues raised and the manuscript can be accepted in its present form.

Response: Thank you.

Reviewer #3 (Remarks to the Author):

The authors have fully addressed all my comments.

Response: Thank you.